# Composite Feature Selection Using Deep Ensembles

**Fergus Imrie**[*]
University of California, Los Angeles
imrie@ucla.edu

**Alexander Norcliffe**[*]
University of Cambridge
alin2@cam.ac.uk

**Pietro Liò**
University of Cambridge
pl219@cam.ac.uk

**Mihaela van der Schaar**
University of Cambridge
The Alan Turing Institute
University of California, Los Angeles
mv472@cam.ac.uk

## Abstract

In many real world problems, features do not act alone but in combination with each other. For example, in genomics, diseases might not be caused by any single mutation but require the presence of multiple mutations. Prior work on feature selection either seeks to identify individual features or can only determine relevant groups from a predefined set. We investigate the problem of discovering groups of predictive features without predefined grouping. To do so, we define predictive groups in terms of linear and non-linear interactions between features. We introduce a novel deep learning architecture that uses an ensemble of feature selection models to find predictive groups, without requiring candidate groups to be provided. The selected groups are sparse and exhibit minimum overlap. Furthermore, we propose a new metric to measure similarity between discovered groups and the ground truth. We demonstrate the utility of our model on multiple synthetic tasks and semi-synthetic chemistry datasets, where the ground truth structure is known, as well as an image dataset and a real-world cancer dataset.

## 1 Introduction

Feature selection is a key problem permeating statistics, machine learning and broader science. Typically in high-dimensional datasets, the majority of features will not be responsible for the target response and thus an important goal is to identify which variables are truly predictive. For example, in healthcare there may be many features (such as age, sex, medical history, etc.) that could be considered, while only a small subset might in fact be relevant for predicting the likelihood of developing a specific disease. By eliminating irrelevant variables, feature selection algorithms can be used to drive discovery, improve model generalisation/robustness, and improve interpretability [18].

However, features often do not act alone but instead in *combination*. In genetics, for instance, it has been noted that understanding the origins of many diseases may require methods able to identify more complex genetic models than single variants [65]. More generally, often there are multiple groups of variables which act (somewhat) independently of each other. For example, in medicine or biology, a number of diseases can manifest from different mechanisms or pathways. Examples include cancer [63], amyotrophic lateral sclerosis (ALS) and frontotemporal dementia (FTD) [6], inflammatory bowel disease [31], cardiovascular disease [42], and diabetes [64].

While feature selection might be able to identify a set of features associated with a particular response, the underlying structure of how features interact is not captured. Further, the resulting

---

[*]Equal contribution

predictive models can be complex, hard to interpret, and not amenable to the generation of hypotheses that can be experimentally tested [46]. This limits the impact such models can have in furthering scientific understanding across many domains where variables are known to interact, such as genetics [65, 69, 60], medicine [89, 13], and economics [8].

Group feature selection is a generalisation of standard feature selection, where instead of selecting individual features, groups of features are either entirely chosen or entirely excluded. A primary application of group feature selection is when features are jointly measured, for example by different instruments. In such scenarios, groups are readily defined as features measured by the same instrument. A natural question is which instruments give the most meaningful measurements. Group feature selection has also been applied in situations where there is extensive domain knowledge regarding the group structure [72] or where groups are defined by the correlation structure between features (e.g. neighbouring pixels in images are highly correlated). The pervasive issue with current group feature selection methods is that a predetermined grouping *must* be provided, and the groups are selected from the given candidates. In reality, we may not know how to group the variables.

In this paper we seek to solve a related but ultimately different and more challenging problem, which we call *Composite Feature Selection*. We wish to find groups of variables *without* prior knowledge, where each group acts as a separate predictive subset of the features and the overall predictive power is greatest when all groups are used in unison. We call each group of features a composite feature.[1] By imposing this structure on the discovered features, we attempt to isolate pathways from features to the response variable. Discovering *groups* of features offers deeper insights into *why* specific features are important than standard feature selection.

**Contributions.** (1) We formalise *composite* feature selection as an extension of standard feature selection, defining composite features in terms of linear and non-linear interactions between variables (Sec. 3). (2) We propose a new deep learning architecture for composite feature selection using an ensemble-based approach (Sec. 4). (3) To assess our solution, we introduce a metric for assessing composite feature similarity based on Jaccard similarity (Sec. 5). (4) We demonstrate the utility of our model on a range of synthetic and semi-synthetic tasks where the ground truth group features are known (Sec. 5). We see that our model not only frequently recovers the relevant features, but also often discovers the underlying group structure. We further illustrate our approach on an image dataset and a real-world cancer dataset, corroborating discovered features and feature interactions in the scientific literature.

## 2   Related Work

Significant attention has been placed on feature selection with a range of solutions including traditional methods (e.g. [56, 47, 34]) and deep learning approaches [54, 91, 7, 49] (see Appendix B for further discussion of standard feature selection). Several approaches have been extended to select predefined groups of variables instead of individual features. For example, LASSO [76, 83] is a linear method that uses an L1 penalty to impose sparsity among coefficients. Group LASSO [93] generalises this to allow predefined groups to be selected or excluded jointly, rather than single features, by replacing the L1 penalty with L2 penalties on each group. Other feature selection methods, such as SLOPE [12], have been similarly extended to group feature selection to give Group-SLOPE [14]. Further examples of group feature selection using adapted loss functions are SCAD-L2 [95] and hierarchical LASSO [100]. Similarly, Bayesian approaches to feature selection [28] have also been generalised to the group setting [35]. Finally, the Knockoff procedure [9, 17, 39, 57, 73, 80] is a generative procedure that creates fake covariates (knockoffs), obeying certain symmetries under permutations of real and knockoff features. By subsequently carrying out feature selection on the combined real and knockoff data, it is possible to obtain guarantees on the False Discovery Rate of the selected features. Generalisations of the Knockoff procedure to the group setting also exist [23, 101], where symmetries under permutations of entire groups must exist.

The key commonality is that none of these methods *discover* groups, but instead can only *select* groups from a set of predefined candidates. Therefore, while they may be applicable when we can split inputs into groups, they are not able to find groups of predictors on their own. Our work differs from these methods by considering the challenge of finding such groups in the absence of prior knowledge. Additionally, unlike prior work, we do not make assumptions about correlations

---

[1]We will often refer to composite features as groups for brevity; in this paper, they refer to the same thing.

between features or place restrictions on groups, such as requiring the candidate groups to partition the features.

# 3 Problem Description

Let $\mathbf{X} \in \mathcal{X}^p$ be a $p$-dimensional signal (such as gene expressions or patient covariates) and $Y \in \mathcal{Y}$ be a response (such as disease traits). Informally, we wish to group features into the maximum number of subsets, $\mathcal{G}_i \subset [p]$, where the predictive power of any single group significantly decreases when any feature is removed, allowing us to separate the groups into different pathways from the signal to the response. Note that we do not enforce assumptions on the groups such as non-overlapping groups or every feature being in at least one group. In this section, we begin with a description of traditional feature selection before formalizing composite feature selection.

## 3.1 Feature Selection

The goal of traditional feature selection is to select a subset, $\mathcal{S} \subset [p]$, of features that are relevant for predicting the response variable. In particular, in the case of embedded feature selection [32], this is conducted jointly with the model selection process.

Let $*$ denote any point not in $\mathcal{X}$ and define $\mathcal{X}_\mathcal{S} = (\mathcal{X} \cup \{*\})^p$. Then, given $\mathbf{X} \in \mathcal{X}^p$, the selected subset of features can be denoted as $\mathbf{X}_\mathcal{S} \in \mathcal{X}_\mathcal{S}$ where $x_{\mathcal{S},k} = x_k$ if $k \in \mathcal{S}$ and $x_{\mathcal{S},k} = *$ if $k \notin \mathcal{S}$. Let $f : \mathcal{X}_\mathcal{S} \to \mathcal{Y}$ be a function in some space $\mathcal{F}$ (such as the space of neural networks) taking subset $\mathbf{X}_\mathcal{S}$ as input to yield $Y$. Then, selecting relevant features for predicting a response can be achieved by solving the following optimization problem:

$$\underset{f \in \mathcal{F},\, \mathcal{S} \subset [p]}{\text{minimize}} \quad \mathbb{E}_{\mathbf{x},y \sim p_{XY}} \Big[ \ell_Y \big( y, f(\mathbf{x}_\mathcal{S}) \big) \Big] \quad \text{subject to} \ \ |\mathcal{S}| \leq \delta, \tag{1}$$

where $\delta$ constrains the number of selected features and $\ell_Y(y, y')$ is a task-specific loss function.

This can be solved by introducing a selection vector $\mathbf{M} = (M_1, \cdots, M_p) \in \{0,1\}^p$, consisting of binary random variables governed by distribution $p_M$, with realization $\mathbf{m}$ indicating selection of the corresponding features. Then, the selected features given vector $\mathbf{m}$ can be written as

$$\tilde{\mathbf{x}} \triangleq \mathbf{m} \odot \mathbf{x} + (1 - \mathbf{m}) \odot \hat{\mathbf{x}}, \tag{2}$$

where $\odot$ indicates element-wise multiplication and $\hat{x}$ are the values assigned to features that are not selected (typically $\hat{\mathbf{x}} \equiv 0$ or $\bar{\mathbf{x}}$). Eq. (1) can be (approximately) solved by jointly learning the model $f$ and the selection vector distribution $p_M$ based on the following optimization problem:

$$\underset{f,\, p_M}{\text{minimize}} \ \mathbb{E}_{\mathbf{x},y \sim p_{XY}} \mathbb{E}_{\mathbf{m} \sim p_M} \Big[ \ell_Y \big( y, f(\tilde{\mathbf{x}}) \big) + \beta \|\mathbf{m}\|_0 \Big], \tag{3}$$

where $\beta$ is a balancing coefficient that controls the number of features to be selected.

## 3.2 Composite Feature Selection

The goal of composite feature selection is not only to find the predictive features, but also to group them based on *how* they are predictive. For example, assume features $x_1$ and $x_2$ are only predictive when both are known by the model, but have the same influence on the outcome independent of $x_3$. Then we wish to group $x_1, x_2$ separately from $x_3$. In this section, we define the embedded composite feature selection problem; that is, we want to find a valid model $f$ and groups $\{\mathcal{G}_1, \ldots, \mathcal{G}_N\}$ in parallel. A model is only valid when the group representations are combined in a way where we can view each group as contributing an independent piece of information for the final prediction. A valid model acts on a *set* of groups [94], thus when combining groups, we require order not to matter. Therefore, we must combine the representations using a permutation invariant aggregator.

Let $A : (\prod_i \mathbb{R}^n) \to \mathbb{R}^N$ be a general permutation invariant aggregation function. It is well established that for a specific choice of $\phi : \mathbb{R}^n \to \mathbb{R}^m$ and $\rho : \mathbb{R}^m \to \mathbb{R}^N$, $A$ can be decomposed as $\rho(\sum_i \phi(\cdot))$ (see [94] for examples). This gives $f(\mathbf{x}) = g\big(\rho\big(\sum_i \phi(f_i(\mathbf{x}_{\mathcal{G}_i}))\big)\big)$, where $f_i$ encodes group $i$, $\rho$ and $\phi$ give the permutation invariant aggregation, and $g$ is any final non-linear function, for instance softmax. The function composition of $\phi$ and $f_i$ can be relabelled as $\tilde{f}_i = \phi \circ f_i$, and the composition of $g$ and $\rho$ can be relabelled as $\tilde{\rho} = g \circ \rho$. This leads to $f(\mathbf{x}) = \tilde{\rho}\big(\sum_i \tilde{f}_i(\mathbf{x}_{\mathcal{G}_i})\big)$, giving the following definition for a valid model structure in composite feature selection.

**Definition 3.1.** The most general valid model for acting on $N$ composite features is given by:

$$f(\mathbf{x}) = \rho\bigg( \sum_{i=1}^{N} f_i(\mathbf{x}_{\mathcal{G}_i}) \bigg). \tag{4}$$

That is, the groups must interact exactly once, all groups must be included, and the interaction is a summation; all other interactions can (and often should) be non-linear.

Depending on the task, a specific permutation invariant aggregation may be chosen (e.g. Max()). However, any permutation invariant aggregator can be (approximately) expressed in the form of Def. 3.1; thus, when learning from data, the general structure of Def. 3.1 means that this is not necessary.

The embedded composite feature selection problem can now be phrased in an analogous way to traditional feature selection. Let $*$ denote some point not in $\mathcal{X}$ and define $\mathcal{X}_{\mathcal{G}_i} = (\mathcal{X} \cup \{*\})^p$. Then, given $\mathbf{X} \in \mathcal{X}^p$, the selected group of features is denoted as $\mathbf{X}_{\mathcal{G}_i} \in \mathcal{X}_{\mathcal{G}_i}$ where $x_{\mathcal{G}_i,k} = x_k$ if $k \in \mathcal{G}_i$ and $x_k = *$ if $k \notin \mathcal{G}_i$. Let $f_i : \mathcal{X}_{\mathcal{G}_i} \to \mathcal{Z}$ be a function in $\mathcal{F}$ that takes as input the subset $\mathbf{X}_{\mathcal{G}_i}$ and outputs a latent representation $\mathbf{z}_i$. Then, finding the groups of features can be achieved by solving the optimization problem:

$$\underset{\rho, f_i \in \mathcal{F}, \, \mathcal{G}_i \subset [p]}{\text{minimize}} \quad \mathbb{E}_{\mathbf{x},y \sim p_{XY}} \left[ \ell_Y \bigg( y, \rho\big( \sum_{i=1}^{N} f_i(\mathbf{x}_{\mathcal{G}_i}) \big) \bigg) \right] \qquad \text{subject to} \quad \begin{array}{l} |\mathcal{G}_i| \leq \delta_i \;\; \forall i, \\ N \geq \Delta, \end{array} \tag{5}$$

where $\boldsymbol{\delta}$ constrains the number of selected features in each group and $\Delta$ gives the minimum number of groups. This objective leads to multiple smaller groups, rather than one group containing all features, which is consistent with our motivation of the problem.

Continuing to expand from traditional feature selection, we can also extend the solution to the composite setting. For $N$ groups we can introduce a selection *matrix* $\mathbf{M} \in \{0, 1\}^{N \times p}$, governed by distribution $p_M$. For a realization $\mathbf{M}$, the selected features from group $i$ are given by

$$\tilde{\mathbf{x}}_i \triangleq \mathbf{m}_i \odot \mathbf{x} + (1 - \mathbf{m}_i) \odot \hat{\mathbf{x}}, \tag{6}$$

where $\mathbf{m}_i$ is the $i^{\text{th}}$ row of $\mathbf{M}$. We can approximately solve Eq. (5) by solving the optimization problem:

$$\underset{f, \, p_M}{\text{minimize}} \;\; \mathbb{E}_{\mathbf{x},y \sim p_{XY}} \mathbb{E}_{\mathbf{M} \sim p_M} \left[ \ell_Y \big( y, f(\mathbf{x}) \big) + R_e(\mathbf{M}) \right], \tag{7}$$

where $f(\mathbf{x})$ obeys Def. (4) and $R_e$ is a regularisation term which controls how features are selected in each group. $R_e$ should capture both group size (i.e. encourage as few features as possible to be selected) but also the relationships between groups (i.e. groups should be distinct and not redundant).

### 3.3 Challenges

There are various challenges in solving the composite feature selection problem. While the ultimate task is to find predictive groups of features, there first remains the necessity simply to identify predictive features, which is already an NP-hard problem [2]. Composite feature selection not only inherits this property but introduces additional complexity since we can think of each group as solving a separate feature selection problem. Consider the number of potential solutions: in traditional feature selection (assuming not all features are selected), there are $2^n - 2$ ways of selecting a subset from $n$ features; even restricting to at most $m << n$ quickly becomes unfeasible for even modest values of $m$. In composite feature selection, *every group* has the same number of solutions as traditional feature selection, drastically increasing the total number of possible solutions. A challenge specific to composite feature selection arises when the ground truth group structure contains groups with overlapping features (e.g. feature $x_1$ interacts independently with both $x_2$ and $x_3$). In this scenario, it is difficult to separate these two effects while penalizing the inclusion of additional features.

## 4   Method: CompFS

In this section, we propose a novel architecture for finding predictive groups of features, which we refer to as **Comp**osite **F**eature **S**election (CompFS). In order to discover groups of features, our model is composed of a set of group selection models and an aggregate predictor. Our approach resembles

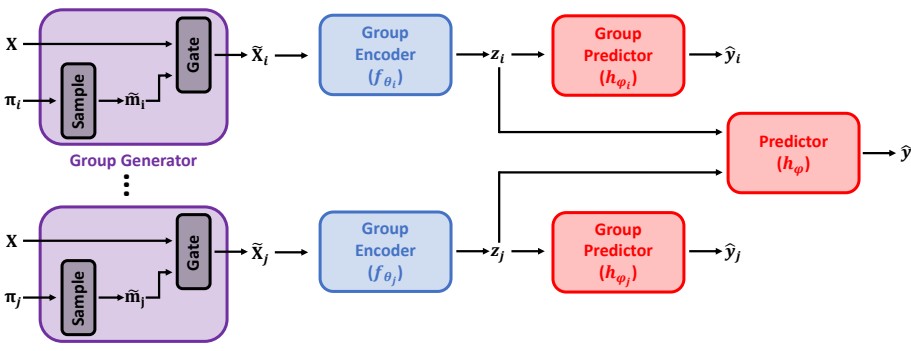

Figure 1: An illustration of CompFS. We use an ensemble of group selection models to discover composite features and an aggregate predictor to combine these features when issuing predictions.

an ensemble of "weak" feature selection models, where each learner attempts to solve the task using a sparse set of features (Figure 1). These models are then trained in such a way as to discover distinct predictive groups. We first consider the group selection models in more detail before describing how the group selection models are combined and the training procedure.

## 4.1 Group Selection Models

CompFS is composed of a set of group selection models, each of which primarily aims to solve the traditional feature selection problem specified by Eq. (1). We achieve this by solving Eq. (3) using a neural network-based approach with stochastic gating of the input features. Each group selection model consists of the following three components (Figure 1):

- *Group Selection Probability*, $\boldsymbol{\pi}_i = (\pi_{1,i}, \cdots, \pi_{p,i}) \in [0,1]^p$, which is a trainable vector that governs the Bernoulli distribution used to generate the gate vector $\mathbf{m}_i$. Each element of the selection probability $\pi_{k,i}$ indicates the importance of the corresponding feature to the target.
- *Group Encoder*, $f_{\theta_i} : \mathcal{X}^p \to \mathcal{Z}$, that takes as input the selected subset of features $\tilde{\mathbf{x}}_i$ and outputs latent representations $\mathbf{z}_i \in \mathcal{Z}$.
- *Group Predictor*, $h_{\phi_i} : \mathcal{Z} \to \mathcal{Y}$, that takes as input the latent representations of the selected subset of features, $\mathbf{z}_i = f_{\theta_i}(\tilde{\mathbf{x}}_i)$, and outputs predictions on the target outcome.

Solving Eq. (3) directly is not possible since the sampling step has no differentiable inverse. Instead, we use the relaxed Bernoulli distribution [59, 38] and apply the reparameterization trick as follows.

Formally, given selection probability $\boldsymbol{\pi} = (\pi_1, \cdots, \pi_p)$ and independent Uniform$(0, 1)$ random variables $(U_1, \cdots, U_p)$, we can generate a relaxed gate vector $\tilde{\mathbf{m}} = (\tilde{m}_1, \cdots, \tilde{m}_p) \in (0, 1)^p$ based on the following reparameterization trick [59]:

$$\tilde{m}_k = \sigma\Big(\frac{1}{\tau}\big(\log \pi_k - \log(1 - \pi_k) + \log U_k - \log(1 - U_k)\big)\Big), \tag{8}$$

where $\sigma(x) = (1 + \exp(-x))^{-1}$ is the sigmoid function. This relaxation is parameterized by $\boldsymbol{\pi}$ and temperature $\tau \in (0, \infty)$. Further, as $\tau \to 0$, the gate vectors $\tilde{m}_k$ converge to Bernoulli$(\pi_k)$ random variables. Crucially, this is differentiable with respect to $\boldsymbol{\pi}$.

Given group selection probability $\boldsymbol{\pi}_i$, we first sample relaxed Bernoulli random variable $\tilde{\mathbf{m}}_i$ according to Eq. (8) and then use $\tilde{\mathbf{m}}_i$ in a gating procedure to select the group of features. The output of the gate is:

$$\tilde{\mathbf{x}}_i = \text{gate}_i(\mathbf{x}) = \tilde{\mathbf{m}}_i \odot \mathbf{x} + (1 - \tilde{\mathbf{m}}_i) \odot \bar{\mathbf{x}}, \tag{9}$$

where we replace the variables that were not selected by their mean value $\bar{\mathbf{x}}$. The mean is used because in certain tasks a feature having a value of 0 may be particularly meaningful. However, any (arbitrary) value could be used for non-selected features. The gate output $\tilde{\mathbf{x}}_i$ is then fed into the group encoder $f_{\theta_i}$ to yield representation $\mathbf{z}_i = f_{\theta_i}(\tilde{\mathbf{x}}_i)$. This representation is finally passed to the group predictor $h_{\phi_i}$ to produce the prediction for an individual learner, $\hat{y}_i = h_{\phi_i}(\mathbf{z}_i)$.

## 4.2 Group Aggregation

The final component necessary for CompFS is a way to aggregate the individual group selection models. This is achieved via an overall *predictor*, $h_\phi : \mathcal{Z} \to \mathcal{Y}$, that takes as input the set of latent representations $\{\mathbf{z}_1, \ldots, \mathbf{z}_N\}$ produced by the individual learners and outputs predictions on the target outcome. For simplicity, we apply a linear prediction head to the latent representations and use element-wise summation to aggregate. Thus, the prediction of the ensemble is given by:

$$\hat{y} = h_\phi(\{\mathbf{z}_1, \ldots, \mathbf{z}_N\}) = \rho\left( \sum_{i=1}^{N} \mathbf{W}_i \mathbf{z}_i + \mathbf{b}_i \right), \tag{10}$$

where $N$ is the number of members of the ensemble (i.e. the number of groups) and $\rho$ is a suitable transformation (e.g. softmax). Note that by using element-wise summation, our model satisfies Def. (4) for acting on composite features.

## 4.3 Loss Functions

**Group Selection Models.** The individual learners can be trained to perform (traditional) feature selection (Eq. (1)) by minimizing the following loss function:

$$\mathcal{L}_{\mathcal{G}_i} = \mathbb{E}_{\mathbf{x}, y \sim p_{XY}} \left[ \ell_Y\big(y, h_{\phi_i}(f_{\theta_i}(\text{gate}_i(\mathbf{x})))\big) + \beta \langle \boldsymbol{\pi}_i \rangle^2 \right], \tag{11}$$

where $\ell_Y$ is a suitable loss function for the prediction task (e.g. cross-entropy for classification tasks and MSE for regression tasks) and $\beta \geq 0$ balances the two terms. Note the selections probabilities $\boldsymbol{\pi}_i$ are not regularized with the typical L1 penalty. Instead, we apply an L2 penalty to the mean selection probability $\langle \boldsymbol{\pi}_i \rangle$ for each individual learner. This is justified as follows. Recall the optimization problem given by Eq. (5). We desire a solution with the maximal number of predictive groups $N$ while minimizing the number of selected features per group $\sum_{i=1}^{N} |\mathcal{G}_i|$. The standard L1 penalty term does not achieve this goal since adding an additional feature to either group $\mathcal{G}_i$ or $\mathcal{G}_j$ incurs the same penalty. In contrast, the L2 penalty imposed on $\langle \boldsymbol{\pi}_i \rangle$ penalizes adding extra features to already large groups, favoring the construction of smaller groups over larger ones.

**Aggregate Predictor.** The aggregate predictor can be trained jointly with the group selection models by minimizing a standard prediction loss (where $\ell_Y$ is the same as in Eq. (11)):

$$\mathcal{L}_E = \mathbb{E}_{\mathbf{x}, y \sim p_{XY}} \left[ \ell_Y\big(y, h_\phi(\{\mathbf{z}_1, \ldots, \mathbf{z}_n\})\big) \right]. \tag{12}$$

**Additional Regularization.** If we simply apply the losses given by Eqs. (11), (12), there will be limited (or even no) differentiation among the individual learners and the optimal solution would be for each learner to simply solve the traditional feature selection problem (Eq. (1)). This results in all learners selecting the same features, which does not achieve our aim of discovering groups of predictive features. In order to encourage differentiation between the models, we introduce an additional loss that penalizes the selection of the same features in multiple groups:

$$\mathcal{L}_R = \mathbb{E}_{\mathbf{x}, y \sim p_{XY}} \left[ \sum_{i=1}^{N} \sum_{j>i} \boldsymbol{\pi}_i \cdot \boldsymbol{\pi}_j \right]. \tag{13}$$

**Overall Loss.** Combining the above, our overall loss function therefore can be written as follows:

$$\mathcal{L} = \sum_{i=1}^{N} \mathcal{L}_{\mathcal{G}_i} + \beta_E \mathcal{L}_E + \beta_R \mathcal{L}_R, \tag{14}$$

where $\beta_E, \beta_R \geq 0$ are hyperparameters to balance the losses.

Training CompFS with the loss given by Eq. (14) is designed to achieve the following: (1) The overall ensemble network should be a good predictor ($\mathcal{L}_E$). (2) Each individual learner should solve the traditional feature selection problem ($\mathcal{L}_{\mathcal{G}_i}$), which requires the group predictor to be accurate while selecting minimal features. However, the individual learners should not be maximally predictive by

definition (hence why we compare individual group feature selection models to weak learners). (3) Finally, we want the groups to be distinct and thus discourage highly similar groups ($\mathcal{L}_R$). However, note that we do not exclude the possibility of some overlap of features between groups. The model is end-to-end differentiable, so we train with gradient descent.

**Evaluation.** During evaluation, only the gating procedure changes. The way features can be selected is chosen by the user. A standard solution which we adopt in this paper is using a threshold $\lambda$ and computing gate vectors $\mathbf{m}_i$ as follows: $m_{i,k} = 1,$ if $\pi_{i,k} > \lambda$ and 0 otherwise.

# 5 Experiments

We evaluate CompFS using several synthetic and semi-synthetic datasets where ground truth feature importances and group structure are known. In addition, we illustrate our method on an image dataset (MNIST) and a real-world cancer dataset (METABRIC). Specific architectural details are given in App. C. Additional information regarding experiments, benchmarks, and datasets can be found in App. D. Additional ablations and sensitivity analysis are in App. A. The code for our method and experiments is available on Github. [2] [3]

**Benchmarks.** The primary goal of our experiments is to demonstrate the utility of discovering composite features over traditional feature selection. Our main benchmark is an oracle feature selection method ("Oracle") that perfectly selects the ground truth features but provides no structure, giving all features as one group. By definition, this is the strongest standard feature selection baseline for the scenarios where the ground truth features are known. We also include comparisons to a linear feature selection method (LASSO) [83] and two non-linear, state of the art approaches, Stochastic Gates (STG) [91] and Supervised Concrete Autoencoder (Sup-CAE) [7]. Finally, we compare with Group LASSO [93], where we enumerate all groups with 1 or 2 features as predefined groups. Note this represents a significant simplification of the task for Group Lasso. We include additional baselines in App. G.

**Metrics.** When the ground truth feature groups $\mathcal{G}_1, \ldots, \mathcal{G}_N$ are known, we use True Positive Rate (TPR) and False Discovery Rate (FDR) to assess the discovered features against the ground truth. To assess composite features, i.e. grouping, we define the Group Similarity ($\mathrm{G_{sim}}$) as the normalized Jaccard similarity between ground truth feature groups and the most similar proposed group:

$$\mathrm{G_{sim}} = \frac{1}{\max(N, K)} \sum_{i=1}^{N} \max_{j \in [K]} \mathcal{J}(\mathcal{G}_i, \hat{\mathcal{G}}_j), \tag{15}$$

where $\mathcal{J}$ is the Jaccard index [37] and $\hat{\mathcal{G}}_1, \ldots, \hat{\mathcal{G}}_K$ are the discovered groups. $\mathrm{G_{sim}} \in [0, 1]$, where $\mathrm{G_{sim}} = 1$ corresponds to perfect recovery of the ground truth groups, while $\mathrm{G_{sim}} = 0$ when none of the correct features are discovered (see App. E for additional details together with examples). We assess the models by seeing if the ground truth features have been correctly discovered, using TPR and FDR. We then see if the underlying grouping has been uncovered (and correct features) using $\mathrm{G_{sim}}$. Finally, we assess the predictive power of the discovered features using accuracy or area under the receiver operating curve (AUROC).

## 5.1 Synthetic Experiments.

**Dataset Description.** We begin by evaluating our method on a range of synthetic datasets where the ground truth feature importance is known (Table 1). We generate synthetic datasets by sampling from the Gaussian distribution with initially no correlations among features ($X \sim \mathcal{N}(0, I)$). We construct binary classification tasks, where the class $y$ is determined by the following decision rules:

- **(Syn1)** $y = 1$ if $x_1 > 0.55$ or $x_2 > 0.55$, 0 otherwise. The ground truth groups are $\{\{1\}, \{2\}\}$. This task assesses whether the model can separate two features rather than group them together.
- **(Syn2)** $y = 1$ if $x_1 x_2 > 0.30$ or $x_3 x_4 > 0.30$, 0 otherwise. The ground truth groups are $\{\{1, 2\}, \{3, 4\}\}$. This task requires identifying groups consisting of more than one variable.
- **(Syn3)** $y = 1$ if $x_1 x_2 > 0.30$ or $x_1 x_3 > 0.30$, 0 otherwise. The ground truth groups are $\{\{1, 2\}, \{1, 3\}\}$. This task investigates whether a model can split the features into two *overlapping* groups of two, rather than one group with all three features.

---

[2] https://github.com/a-norcliffe/Composite-Feature-Selection
[3] https://github.com/vanderschaarlab/Composite-Feature-Selection

Table 1: Performance on Synthetic Datasets, values are recorded with their standard deviations.

| DATASET | MODEL | TPR | FDR | $G_{SIM}$ | NO. GROUPS | ACCURACY (%) |
|---|---|---|---|---|---|---|
| SYN1 | COMPFS(5) | $100.0 \pm 0.0$ | $0.0 \pm 0.0$ | $0.91 \pm 0.14$ | $2.2 \pm 0.4$ | $98.9 \pm 0.5$ |
| | ORACLE | $100.0 \pm 0.0$ | $0.0 \pm 0.0$ | $0.50 \pm 0.00$ | $1.0 \pm 0.0$ | $100.0 \pm 0.0$ |
| | LASSO | $100.0 \pm 0.0$ | $0.0 \pm 0.0$ | $0.50 \pm 0.00$ | $1.0 \pm 0.0$ | $81.8 \pm 2.0$ |
| | GROUP LASSO | $100.0 \pm 0.0$ | $0.0 \pm 0.0$ | $0.67 \pm 0.00$ | $3.0 \pm 0.0$ | $83.8 \pm 1.4$ |
| | STG | $100.0 \pm 0.0$ | $0.0 \pm 0.0$ | $0.50 \pm 0.00$ | $1.0 \pm 0.0$ | $97.8 \pm 1.4$ |
| | SUP-CAE | $100.0 \pm 0.0$ | $0.0 \pm 0.0$ | $0.50 \pm 0.00$ | $1.0 \pm 0.0$ | $97.8 \pm 1.4$ |
| SYN2 | COMPFS(5) | $95.0 \pm 15.0$ | $0.0 \pm 0.0$ | $0.90 \pm 0.20$ | $1.8 \pm 0.4$ | $95.5 \pm 5.4$ |
| | ORACLE | $100.0 \pm 0.0$ | $0.0 \pm 0.0$ | $0.50 \pm 0.00$ | $1.0 \pm 0.0$ | $100.0 \pm 0.0$ |
| | LASSO | $0.0 \pm 0.0$ | $0.0 \pm 0.0$ | $0.00 \pm 0.00$ | $0.0 \pm 0.0$ | $52.6 \pm 2.9$ |
| | GROUP LASSO | $0.0 \pm 0.0$ | $0.0 \pm 0.0$ | $0.00 \pm 0.00$ | $0.0 \pm 0.0$ | $52.2 \pm 0.9$ |
| | STG | $100.0 \pm 0.0$ | $0.0 \pm 0.0$ | $0.50 \pm 0.00$ | $1.0 \pm 0.0$ | $93.9 \pm 2.2$ |
| | SUP-CAE | $37.5 \pm 31.7$ | $42.5 \pm 44.2$ | $0.24 \pm 0.20$ | $1.0 \pm 0.0$ | $61.9 \pm 12.8$ |
| SYN3 | COMPFS(5) | $100.0 \pm 0.0$ | $0.0 \pm 0.0$ | $0.68 \pm 0.05$ | $1.3 \pm 0.5$ | $97.4 \pm 1.1$ |
| | ORACLE | $100.0 \pm 0.0$ | $0.0 \pm 0.0$ | $0.67 \pm 0.00$ | $1.0 \pm 0.0$ | $100.0 \pm 0.0$ |
| | LASSO | $0.0 \pm 0.0$ | $0.0 \pm 0.0$ | $0.00 \pm 0.00$ | $0.0 \pm 0.0$ | $56.5 \pm 4.0$ |
| | GROUP LASSO | $0.0 \pm 0.0$ | $0.0 \pm 0.0$ | $0.00 \pm 0.00$ | $0.0 \pm 0.0$ | $54.6 \pm 1.3$ |
| | STG | $100.0 \pm 0.0$ | $0.0 \pm 0.0$ | $0.67 \pm 0.00$ | $1.0 \pm 0.0$ | $95.3 \pm 1.7$ |
| | SUP-CAE | $23.3 \pm 31.6$ | $66.7 \pm 47.1$ | $0.23 \pm 0.31$ | $1.0 \pm 0.0$ | $62.6 \pm 12.6$ |
| SYN4 | COMPFS(5) | $90.0 \pm 12.2$ | $51.9 \pm 13.8$ | $0.47 \pm 0.20$ | $2.5 \pm 0.7$ | $95.8 \pm 1.8$ |
| | ORACLE | $100.0 \pm 0.0$ | $0.0 \pm 0.0$ | $0.50 \pm 0.00$ | $1.0 \pm 0.0$ | $100.0 \pm 0.0$ |
| | LASSO | $0.0 \pm 0.0$ | $0.0 \pm 0.0$ | $0.00 \pm 0.00$ | $0.0 \pm 0.0$ | $51.8 \pm 3.2$ |
| | GROUP LASSO | $0.0 \pm 0.0$ | $10.0 \pm 31.6$ | $0.00 \pm 0.00$ | $0.1 \pm 0.3$ | $53.0 \pm 1.1$ |
| | STG | $100.0 \pm 0.0$ | $66.7 \pm 0.0$ | $0.17 \pm 0.00$ | $1.0 \pm 0.0$ | $94.2 \pm 2.1$ |
| | SUP-CAE | $72.5 \pm 14.2$ | $16.7 \pm 14.7$ | $0.39 \pm 0.08$ | $1.0 \pm 0.0$ | $72.2 \pm 13.2$ |

- **(Syn4)** $y = 1$ if $x_1 x_4 > 0.30$ or $x_7 x_{10} > 0.30$, 0 otherwise. The ground truth groups are $\{\{1, 4\}, \{7, 10\}\}$. This task is equivalent to **Syn2**, however, here the features exhibit strong correlation in collections of 3, i.e. features 1, 2, and 3 are highly correlated, features 4, 5, and 6 are highly correlated, and so on. This task demonstrates the difficulty of carrying out group feature selection (and indeed standard feature selection) when the features are highly correlated.

The decision rules are created such that there is minimal class imbalance. We use signals with 500 dimensions to demonstrate the utility in the high dimensional regime. We use 20,000 samples to train and 200 to test. Each experiment is repeated 10 times.

**Analysis.** On both Syn1 and Syn2, CompFS achieves high TPR with no false discoveries (0% FDR) and significantly higher $G_{sim}$ than the Oracle. Despite allowing CompFS to discover up to 5 groups, CompFS typically finds the correct number of groups (2), demonstrating that it is not necessary for the number of potential composite features to match the ground truth, which is vital in real-world use cases where this is unknown. Syn3 is significantly more challenging due to the overlapping structure and we observe essentially the same performance as Oracle. Despite finding all the correct features and no false discoveries, CompFS typically finds the union $\{1, 2, 3\}$ rather than the underlying group structure $\{\{1, 2\}, \{1, 3\}\}$. Finally, for Syn4, while CompFS has a relatively high FDR, it frequently finds the ground truth relevant features and groups with similar $G_{sim}$ to Oracle. This is a challenging task with significant correlation between features. Despite this, CompFS is able to uncover the underlying group structure, providing additional insight over traditional feature selection. STG typically performs reasonably in terms of traditional feature selection, but scores poorly in terms of $G_{sim}$ due to not providing any group information.

### 5.2 Semi-Synthetic Experiments.

**Dataset Description.** Next, we assess our ability to identify composite features using semi-synthetic molecular datasets. These tasks are analogs of real-world problems, such as identifying biologically active chemical groups; however, the labels are determined by a synthetic "binding logic" so that the ground truth feature relevance is known. We use several of the datasets constructed by [62], some of which were also used by [75].[4] The synthetic "binding logics" are expressed as a combination of molecular fragments that must either be present or absent for binding to occur and are used to label molecules from the ZINC database [36]. Each logic includes up to four functional groups (Table 6).

---

[4]Data from `https://github.com/google-research/graph-attribution/raw/main/data/all_16_logics_train_and_test.zip`.

Molecules are featurized using a set of 84 functional groups, where feature $x_i = 1$ if the molecule contains functional group $i$ and 0 otherwise. The specific binding logics are given in App. F.

Table 2: Performance on Chemistry Datasets, values are recorded with their standard deviations.

| DATASET | MODEL | TPR | FDR | $G_{SIM}$ | NO. GROUPS | ACCURACY (%) |
|---------|-------|-----|-----|-----------|------------|--------------|
| CHEM1 | COMPFS(5) | $100.0 \pm 0.0$ | $0.0 \pm 0.0$ | $0.82 \pm 0.20$ | $1.9 \pm 0.5$ | $100.0 \pm 0.0$ |
| | ORACLE | $100.0 \pm 0.0$ | $0.0 \pm 0.0$ | $0.50 \pm 0.00$ | $1.0 \pm 0.0$ | $100.0 \pm 0.0$ |
| | LASSO | $100.0 \pm 0.0$ | $0.0 \pm 0.0$ | $0.50 \pm 0.00$ | $1.0 \pm 0.0$ | $75.8 \pm 0.0$ |
| | GROUP LASSO | $100.0 \pm 0.0$ | $0.0 \pm 0.0$ | $0.67 \pm 0.00$ | $3.0 \pm 0.0$ | $100.0 \pm 0.0$ |
| | STG | $100.0 \pm 0.0$ | $0.0 \pm 0.0$ | $0.50 \pm 0.00$ | $1.0 \pm 0.0$ | $100.0 \pm 0.0$ |
| | SUP-CAE | $62.5 \pm 13.2$ | $23.3 \pm 17.5$ | $0.37 \pm 0.07$ | $1.0 \pm 0.0$ | $77.8 \pm 11.0$ |
| CHEM2 | COMPFS(5) | $100.0 \pm 0.0$ | $0.0 \pm 0.0$ | $0.72 \pm 0.24$ | $2.2 \pm 0.6$ | $100.0 \pm 0.0$ |
| | ORACLE | $100.0 \pm 0.0$ | $0.0 \pm 0.0$ | $0.50 \pm 0.00$ | $1.0 \pm 0.0$ | $100.0 \pm 0.0$ |
| | LASSO | $100.0 \pm 0.0$ | $0.0 \pm 0.0$ | $0.50 \pm 0.00$ | $1.0 \pm 0.0$ | $81.6 \pm 0.0$ |
| | GROUP LASSO | $100.0 \pm 0.0$ | $0.0 \pm 0.0$ | $0.40 \pm 0.00$ | $5.0 \pm 0.0$ | $81.6 \pm 0.0$ |
| | STG | $100.0 \pm 0.0$ | $0.0 \pm 0.0$ | $0.50 \pm 0.00$ | $1.0 \pm 0.0$ | $100.0 \pm 0.0$ |
| | SUP-CAE | $66.7 \pm 0.0$ | $0.0 \pm 0.0$ | $0.42 \pm 0.00$ | $1.0 \pm 0.0$ | $80.9 \pm 9.5$ |
| CHEM3 | COMPFS(5) | $100.0 \pm 0.0$ | $7.3 \pm 11.7$ | $0.62 \pm 0.17$ | $2.4 \pm 0.5$ | $100.0 \pm 0.0$ |
| | ORACLE | $100.0 \pm 0.0$ | $0.0 \pm 0.0$ | $0.50 \pm 0.00$ | $1.0 \pm 0.0$ | $100.0 \pm 0.0$ |
| | LASSO | $100.0 \pm 0.0$ | $0.0 \pm 0.0$ | $0.50 \pm 0.00$ | $1.0 \pm 0.0$ | $87.4 \pm 5.2$ |
| | GROUP LASSO | $100.0 \pm 0.0$ | $20.0 \pm 0.0$ | $0.20 \pm 0.00$ | $10.0 \pm 0.0$ | $91.5 \pm 0.0$ |
| | STG | $100.0 \pm 0.0$ | $0.0 \pm 0.0$ | $0.50 \pm 0.00$ | $1.0 \pm 0.0$ | $100.0 \pm 0.0$ |
| | SUP-CAE | $62.5 \pm 13.2$ | $23.3 \pm 17.5$ | $0.37 \pm 0.07$ | $1.0 \pm 0.0$ | $77.8 \pm 11.0$ |

**Analysis.** All methods are able to identify the ground truth relevant features; however, only CompFS provides deeper insights. Unlike for Syn1-4, LASSO correctly selects the ground truth features since the dataset consists of binary variables and thus it is possible to find performant linear models. However, while discovering the correct features, Group LASSO selects all possible combinations of these features, adding no benefit over standard feature selection.

For Chem1-2, CompFS perfectly recovers the group structure in the majority of experiments, leading to high $G_{sim}$ far exceeding traditional feature selection. On Chem3, we occasionally discover additional features that are not part of the binding logic. However, a number of molecular fragments are strongly correlated with the binding logic, even though they are not themselves included. In fact, some features contain information about *multiple* functional groups. For example, esters contain a carbonyl and an ether; both are in the binding logic for Chem3, while ester is not, despite being highly informative, and thus occasionally CompFS incorrectly selects this feature. In spite of this, CompFS achieves significantly higher $G_{sim}$ than even Oracle. This demonstrates the benefit of the grouping discovered by CompFS, even with a modest number of false discoveries. As before, CompFS typically finds the correct number of groups (2), despite being able to discover up to 5 groups, further demonstrating that the number of composite features need not be known *a priori*, which is the case in real-world applications.

### 5.3 MNIST

**Dataset Description.** We investigate CompFS on the MNIST dataset [48]. While this well-known dataset consists of 28x28 images and typically fixed pixel locations do not have specific meaning, it has been extensively used in the feature selection literature due to the handwritten digits being centered and scaled, thus each of the 784 pixels can be (somewhat) meaningfully treated as a separate feature. While the ground truth group structure is unknown, a benefit of MNIST is that we can readily visualise selected features.

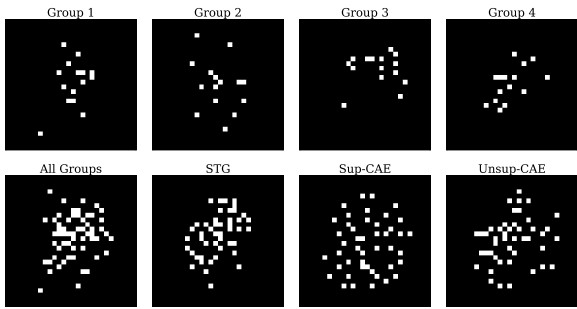

Figure 2: Pixels selected by CompFS and baselines.

**Analysis.** The features discovered by CompFS (using 4 groups), STG, Sup-CAE and Unsup-CAE are shown in Figure 2. As expected, all selected pixels are central and relatively spread out. However, the four groups discovered by CompFS appear to have slightly different focus, in particular Group 3.

To investigate the impact of these differences, we evaluate the predictive power of each of the groups. We find that the individual groups have relatively low accuracies (72%-81%), in part due to only using 15 pixels (<2% of total). However, the union of these features achieve significantly greater accuracy of 95%, equalling the performance of STG and (Sup-,Unsup-)CAE. This illustrates that the grouping does not seem to drastically affect performance, despite enforcing constraints on how the features are used by the model.

Finally, we consider the per-class accuracy of each of these groups. Interestingly, the variance in performance between classes is significantly higher for the groups than any of the overall methods (Fig. 4). For example, Group 2 struggles to identify digits 4 and 5, while Group 3 performs poorly on digits 2, 3, and 8. This highlights the distinct information contained in each group. Further details and analysis is provided in App. G.3.

### 5.4 Real-World Data: METABRIC

**Dataset Description.** Finally, we assess CompFS on a real-world dataset, METABRIC [21, 68], where the ground truth group structure is *unknown*. METABRIC contains gene expression, mutation, and clinical data for 1,980 primary breast cancer samples. We evaluated the ability to predict the progesterone receptor (PR) status of the tissue based on the gene expression data, which consists of measurements for 489 genes.

Table 3: METABRIC performance. We compare CompFS and STG using 25 features to an MLP using all 489 features.

| Model | AUROC |
|---|---|
| MLP (All features) | 0.869 |
| CompFS(5) | 0.830 |
| STG | 0.843 |

**Analysis.** CompFS suffers limited performance degradation compared to using all features, despite only using 5% of the features (Table 3). Despite imposing a more rigid structural form on how features can interact in the predictive model, STG only had marginally greater predictive power than CompFS. However, CompFS provides greater insight into how the features interact than STG.

We found supporting evidence in the scientific literature for all but 1 of the genes discovered by CompFS (Table 10). In addition, within each group, we found further evidence of the interactions between genes, demonstrating the ability for CompFS to learn informative groups of features. For example, in Group 1, CXCR1 and PEN-2 (the protein encoded by PSENEN) are known to interact [5]. In Group 2, BMP6 encodes a member of the TGF-$\beta$ superfamily of proteins, and TGF-$\beta$ triggers activation of SMAD3 [19]. In the same group, MAPK1 activity is dependent on the activity of PRKCQ in breast cancer cells [15], while MAPK1 is also known to interact with MAPT [51], SMAD3 [26], and BMP6 [96]. Additional supporting evidence can be found in Appendix H.

## 6 Conclusion

In this paper, we introduced CompFS, an ensemble-based approach that tackles the newly proposed challenge of composite feature selection. Using synthetic and semi-synthetic data, we assess our ability to go beyond traditional feature selection and recover deeper underlying connections between variables. CompFS is not without limitations: as with other methods, points of difficulty arise when features are highly correlated, or if predictive composites contain overlapping features. Future work may overcome this by using correlated gates. Further, as with many traditional feature selection methods, there are no guarantees on false discovery rate. This could be tackled by first proposing candidate composite features, and then using the Group Knockoff procedure. Additionally, to discover groups, CompFS requires the introduction of additional hyperparameters which could be challenging to tune in practice. More broadly, as with standard feature selection, groups found under composite feature selection must be verified by domain experts (both features but additionally interactions). However, we believe the additional structure provided by composite feature selection could be of significant benefit to a wide variety of practitioners.

## Acknowledgements

We thank the anonymous reviewers for their comments and suggestions. We also thank Bogdan Cebere and Evgeny Saveliev for reviewing our public code. Fergus Imrie and Mihaela van der Schaar are supported by the National Science Foundation (NSF, grant number 1722516). Mihaela van der Schaar is additionally supported by the Office of Naval Research (ONR). Alexander Norcliffe is supported by a GlaxoSmithKline grant.

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
