# OpenReview forum: "Composite Feature Selection Using Deep Ensembles"
_NeurIPS.cc/2022/Conference — NeurIPS 2022 Accept_

### Official Review · Reviewer_iDHM · 2022-07-05

**Rating:** 6
**Confidence:** 3
**Soundness:** 3 good
**Presentation:** 3 good
**Contribution:** 2 fair

**Summary:**

This paper proposed a composite feature selection, which aims to find groups of features with combined effects for prediction. More specifically, the authors proposed a deep learning architecture that uses an ensemble of feature selection models to find predictive groups. Experimental results on synthetic datasets, semi synthetic datasets and MNIST dataset demonstrated the effectiveness of the proposed method.

**Questions:**

1. Can you use random forest or GBDT for feature grouping? For example, you can use the features shown in the same tree as a group.
2. For feature grouping, a trivial way is to cluster the features with clustering methods. How does the proposed method perform compared with clustering methods, e.g., spectrum clustering?


**Limitations:**

The authors have adequately addressed the limitations and potential negative societal impact of their work.

**Strengths And Weaknesses:**

Pros
1. The paper proposed a new setting for feature selection problems, i.e., selecting feature groups that features in each group may interact with each other.
2. The experimental results seem promising.
3. This paper is well-written and easy to follow.

Cons:
1. The problem setting seems to be related to some existing works. It seems to me that the setting in this paper is similar to the sub structures of the many machine learning tasks, e.g., there are several sub tasks which may or may not overlap with each other. In this paper, the authors proposed a solution for automatically discovering these sub tasks with the goals that all sub tasks can perform well and the ensemble of them can achieve optimal performance. This might be related to traditional ensemble learning methods, e.g., random forest or GBDT, in which many trees (upon a subset of features) are learned individually and then combined to achieve optimal accuracy.
2. In the abstract, it is mentioned that “For example, in genomics, diseases might not be caused by any single mutation but require the presence of multiple mutations”. This may not be supportive to the motivation of this paper. It might be better to show some examples of how different combinations of mutations can cause the same disease. But I am not sure how often it is in real diseases.
3. Section 3 mentioned the challenge that different groups may have overlapping features, but I did not see how the proposed method can address such challenge.
4. The experiments on semi-synthetic datasets seem interesting, but it could be more convincing if there were some experiments on real medical/chemical datasets.

---

> ### Author Response · Authors · 2022-08-02
> **Response to iDHM (3/3)**
>
> __Motivation__
>
> The example given in the abstract was simply meant to highlight that methods selecting individual features as independent predictors are insufficient for tackling many problems.
>
> Indeed, the situation the reviewer describes is very common and many diseases can be caused through different pathways. A number of examples in medicine/biology can be found, such as cancer (Meira et al., 2001), amyotrophic lateral sclerosis (ALS) and frontotemporal dementia (FTD) (Balendra and Isaacs, 2018), inflammatory bowel disease (Graham and Xavier, 2020), cardiovascular disease (Kelly and Semsarian, 2009), and diabetes (Merino et al., 2022).
>
> __Sub tasks__
>
> We find the parallel between composite feature selection and sub-tasks interesting and think this could be further explored in future work.
>
> Regarding the relationship with traditional ensemble learning methods, see "Additional Baselines - RF / GBDT" (above in our reply) for a comparison of CompFS with baselines based on random forests and gradient boosted decision trees.
>
> __Overlapping features__
>
> We agree that our method is not specifically designed to handle overlapping features, although CompFS has the flexibility to select the same feature in two separate groups. We leave this as a challenge for future work and have highlighted this point more clearly as a limitation of our work (see Sec 6. Conclusion).
>
> __References__
>
> 1. R. Balendra and A. M. Isaacs. _C9orf72_-mediated ALS and FTD: multiple pathways to disease, _Nature Reviews Neurology_ 14:544–558 (2018).
> 2. D. B. Graham and R. J. Xavier. Pathway paradigms revealed from the genetics of inflammatory bowel disease. _Nature Reviews Neurology_ 14:544–558 (2018).
> 3. M. Kelly and C. Semsarian. Multiple Mutations in Genetic Cardiovascular Disease: A Marker of Disease Severity? _Advances in Genetics, Proteomics, and Metabolomics_ 2:182-190 (2009).
> 4. L. B. Meira et al. Cancer predisposition in mutant mice defective in multiple genetic pathways: uncovering important genetic interactions. _Mutation Research/Fundamental and Molecular Mechanisms of Mutagenesis_ 477(1-2):51-58 (2001).
> 5. J. Merino et al. Polygenic scores, diet quality, and type 2 diabetes risk: An observational study among 35,759 adults from 3 US cohorts. _PLOS Medicine_ 19(4):e1003972 (2022)
> 6. C. Curtis et al. The genomic and transcriptomic architecture of 2,000 breast tumours reveals novel subgroups. _Nature_ 486(7403):346–352, (2012).
> 7. B. Pereira et al. The somatic mutation profiles of 2,433 breast cancers refine their genomic and transcriptomic landscapes. _Nature Communications_ 7(1):11479 (2016).
> 8. M. Bakele et al. An interactive network of elastase, secretases, and PAR-2 protein regulates CXCR1 receptor surface expression on neutrophils. _Journal of Biological Chemistry_, 289(30):20516–20525 (2014).
> 9. B. Chen et al. Differential effects of smad2 and smad3 in regulation of macrophage phenotype and function in the infarcted myocardium. _Journal of Molecular and Cellular Cardiology_, 171:1–15 (2022).
> 10. J. Byerly et al. PRKCQ promotes oncogenic growth and anoikis resistance of a subset of triple-negative breast cancer cells. _Breast Cancer Research_, 18(1):95 (2016).
> 11. C. Leugers et al. Tau in MAPK activation. _Frontiers in Neurology_, 4, (2013).
> 12. W. B. Fang et al. CCL2/CCR2 chemokine signaling coordinates survival and motility of breast cancer cells through smad3 protein- and p42/44 mitogen-activated protein kinase (MAPK)-dependent mechanisms. _Journal of Biological Chemistry_, 287(43):36593–36608 (2012).
> 13. X.-Y. Zhang et al. BMP6 downregulates GDNF expression through SMAD1/5 and ERK1/2 signaling pathways in human granulosa-lutein cells. _Endocrinology_, 159(8):2926–2938, (2018).

---

> > ### Comment · Reviewer_iDHM · 2022-08-07
> > **Thanks for your response**
> >
> > I think the authors have addressed most of my concerns in the rebuttal. I would like to increse my score to 6.

---

> > > ### Author Response · Authors · 2022-08-07
> > > **Thank you for your response**
> > >
> > > Dear reviewer iDHM,
> > >
> > > Thank you for your response and for raising your score. We appreciate the effort in helping improve the paper.
> > >
> > > If there are any outstanding points you would like to discuss, we would be more than happy to address them.

---

> ### Author Response · Authors · 2022-08-02
> **Response to iDHM (2/3)**
>
> __Additional baselines__
>
> _Random forest / Gradient boosted decision trees:_
>
> We implemented two additional baselines using random forests (RF) and gradient boosted decision trees (GBDT). We selected the $n$ trees with the best predictive performance on the validation set ($n=10$) and selected features based on the feature importances for each tree. Features were selected if the feature importance exceeded a threshold $\lambda$ ($\lambda=0.1$) and we discarded duplicate groups. Results can be seen below:
>
> Synthetic datasets:
>
> | Dataset | Method | TPR   | FDR  | G$_{\text{sim}}$ | No. Groups | Accuracy (%) |
> |:-------:|:------:|:-----:|:----:|:-----:|:----------:|:------------:|
> | SYN1    | CompFS | 100.0 | 0.0  | 0.91  | 2.2        | 98.9         |
> |         | RF     | 100.0 | 0.0  | 0.68  | 2.8        | 96.6         |
> |         | GBDT   | 100.0 | 0.0  | 0.50  | 1.0        | 100.0        |
> | SYN2    | CompFS | 95.0  | 0.0  | 0.90  | 1.8        | 95.5         |
> |         | RF     | 100.0 | 0.0  | 0.28  | 6.8        | 62.9         |
> |         | GBDT   | 100.0 | 0.0  | 0.54  | 2.2        | 99.0         |
> | SYN3    | CompFS | 100.0 | 0.0  | 0.68  | 1.3        | 97.4         |
> |         | RF     | 100.0 | 0.0  | 0.54  | 3.6        | 51.3         |
> |         | GBDT   | 100.0 | 0.0  | 0.67  | 1.0        | 98.5         |
> | SYN4    | CompFS | 90.0  | 51.9 | 0.47  | 2.5        | 95.8         |
> |         | RF     | 100.0 | 63.5 | 0.12  | 9.8        | 82.1         |
> |         | GBDT   | 100.0 | 0.0  | 0.44  | 3.0        | 98.5         |
>
> Chemistry datasets:
>
> | Dataset | Method | TPR   | FDR  | G$_{\text{sim}}$ | No. Groups | Accuracy (%) |
> |:-------:|:------:|:-----:|:----:|:-----:|:----------:|:------------:|
> | CHEM1   | CompFS | 100.0 | 0.0  | 0.82  | 1.9        | 100.0        |
> |         | RF     | 100.0 | 20.0 | 0.50  | 1.6        | 98.4         |
> |         | GBDT   | 100.0 | 0.0  | 0.50  | 1.0        | 100.0        |
> | CHEM2   | CompFS | 100.0 | 0.0  | 0.72  | 2.2        | 100.0        |
> |         | RF     | 100.0 | 25.0 | 0.47  | 2.4        | 85.1         |
> |         | GBDT   | 100.0 | 0.0  | 0.50  | 1.0        | 100.0        |
> | CHEM3   | CompFS | 100.0 | 7.3  | 0.62  | 2.4        | 100.0        |
> |         | RF     | 100.0 | 46.8 | 0.15  | 8.0        | 95.9         |
> |        | GBDT   | 100.0 | 0.0  | 0.50  | 1.0        | 100.0        |
>
> For all datasets, CompFS outperforms both RF and GBDT in terms of G$\_{\text{sim}}$. The random forest performed poorly with high FDR and low G$\_{\text{sim}}$, identifying many additional groups. GBDT displayed perfect TPR and FDR across the datasets, but was not able to effectively group features, typically identifying only a single group and thus achieving lower G$_{\text{sim}}$ than CompFS.
>
> _Clustering:_
>
> As suggested by the reviewer, we explored the effect of clustering features. We used agglomerative feature clustering, a type of hierarchical clustering, to cluster the correct, ground truth features. In addition, we used the correct number of groups as the number of clusters. Both of these factors make the task significantly easier and are not representative of real world use.
>
> However, despite these advantages, clustering the ground truth features did not recover the ground truth groups. This is due to the clustering algorithm being based on correlations between features rather than their functional relationship with the target. As a result, the G$_\text{sim}$ was often substantially lower than CompFS. Note the outperformance of clustering on Syn4 is almost entirely due to only considering ground truth features, while CompFS has non-zero FDR.
>
> Synthetic datasets:
>
> | Dataset | Method | TPR   | FDR  | G$_{\text{sim}}$ | No. Groups | Accuracy (%) |
> |:-------:|:------:|:-----:|:----:|:-----:|:----------:|:------------:|
> | SYN1    | CompFS | 100.0 | 0.0  | 0.91  | 2.2        | 98.9         |
> |      | Cluster| 100.0 | 0.0  | 1.00  | 2.0        | -            |
> | SYN2    | CompFS | 95.0  | 0.0  | 0.90  | 1.8        | 95.5         |
> |      | Cluster| 100.0 | 0.0  | 0.58  | 2.0        | -            |
> | SYN3    | CompFS | 100.0 | 0.0  | 0.68  | 1.3        | 97.4         |
> |      | Cluster| 100.0 | 0.0  | 0.50  | 2.0        | -            |
> | SYN4    | CompFS | 90.0  | 51.9 | 0.47  | 2.5        | 95.8         |
> |      | Cluster| 100.0 | 0.0  | 0.58  | 2.0        | -            |
>
> Chemistry datasets:
>
> | Dataset | Method | TPR   | FDR  | G$_{\text{sim}}$ | No. Groups | Accuracy (%) |
> |:-------:|:------:|:-----:|:----:|:-----:|:----------:|:------------:|
> | CHEM1   | CompFS | 100.0 | 0.0  | 0.82  | 1.9        | 100.0        |
> |         | Cluster| 100.0 | 0.0  | 1.00  | 2.0        | -            |
> | CHEM2   | CompFS | 100.0 | 0.0  | 0.72  | 2.2        | 100.0        |
> |         | Cluster| 100.0 | 0.0  | 0.50  | 2.0        | -            |
> | CHEM3   | CompFS | 100.0 | 7.3  | 0.62  | 2.4        | 100.0        |
> |         | Cluster| 100.0 | 0.0  | 0.58  | 2.0        | -            |

---

> ### Author Response · Authors · 2022-08-02
> **Response to iDHM (1/3)**
>
> *Dear reviewer iDHM, thank you for the feedback on our work. We have addressed your comments in the updated version of our manuscript and provide a point-by-point response below.*
>
> __Additional dataset: Real world cancer dataset__
>
> We have tested CompFS on the METABRIC dataset (Curtis et al., 2012; Pereira et al., 2016). METABRIC contains gene expression, mutation, and clinical data for 1,980 primary breast cancer samples. We evaluated the ability to predict the progesterone receptor (PR) status of the tissue based on the gene expression data, which consists of measurements for 489 genes.
>
> Prediction performance:
>
> | Method             | Accuracy | AUC ROC | # Features |
> |--------------------|----------|---------|------------|
> | MLP (All features) | 77.95%   | 0.869   | 489        |
> | CompFS             | 75.07%   | 0.830   | 25         |
> | STG                | 76.12%   | 0.843   | 25         |
> | CAE                | 76.64%   | 0.856   | 25         |
>
> CompFS performs strongly, with limited performance degradation compared to using all features, despite only using 5% of the total features.
> Despite imposing a more rigid structural form on the predictive model and how features can interact, STG and CAE only had marginally greater predictive power than CompFS. However, CompFS provides greater insight into how the features interact (see below).
>
> Qualitative assessment:
>
> We also examined the groups of features selected by CompFS. We found supporting evidence in the scientific literature for all but 1 of the discovered genes (Table 10). In addition, within each group, we found further evidence of the interactions between genes, demonstrating the ability for CompFS to learn informative groups of features.
>
> For example, in Group 1, CXCR1 and PEN-2 (the protein encoded by PSENEN) are known to interact (Bakele et al., 2014).  In Group 2, BMP6 encodes a member of the TGF-$\beta$, and TGF-$\beta$ triggers activation of SMAD3 (Chen et al., 2022). In the same group, MAPK1 activity is dependent on the activity of PRKCQ in breast cancer cells (Byerly et al., 2016), while MAPK1 is also known to interact with MAPT (Leugers et al., 2013), SMAD3 (Fang et al., 2012), and BMP6 (Zhang et al., 2018).
>
> We have included additional supporting evidence in Section 5.3 and Appendix H. Selected features and groups are shown below.
>
> | Group | Features |
> |:----:|:----:|
> | Group 1 | psenen, cxcr1, dlec1, mmp15, srd5a3 |
> | Group 2 | bmp6, mapk1, smad3, mapt, prkcq |
> | Group 3 | bmpr2, mmp12, asxl2, birc6, star |
> | Group 4 | cdkn1a, fgfr1, tgfbr3, npnt, akr1c3 |
> | Group 5 | bmp4, mmp10, tbl1xr1, ush2a, hsd17b1 |

---

> ### Author Response · Authors · 2022-08-05
> **Dear Reviewer iDHM**
>
> Dear Reviewer iDHM,
>
> Once again, we would like to thank you for your feedback on our work! We hope that our rebuttal has addressed any questions or concerns you may have had about our paper. If you have any other comments or concerns, please let us know - we would be happy to do our utmost to address them during the author-reviewer discussion period (which ends this Tuesday)!
>
> Thank you.

---

### Official Review · Reviewer_LHmD · 2022-07-06

**Rating:** 5
**Confidence:** 4
**Soundness:** 2 fair
**Presentation:** 2 fair
**Contribution:** 2 fair

**Summary:**

Most features of a sample are usually not independent but interdependent. In this paper, the author(s) introduced a method combining deep learning and ensemble learning to identify informative feature groups. Empirical experiments were implemented to validate the performance of the proposed method.

**Questions:**

As for the Weaknesses mentioned above, to be specific, I have the following questions/comments which need further explanation from the author(s). Please correct me if I am wrong. Thanks.

1. In lines 10-11, it mentioned that "...we can rank the groups based on their predictive power for downstream decision making..." I didn't see an empirical example of this, or did I miss it?

2. In lines 46-47, "...why specific features are important than standard feature selection, with the goal of improving data-driven...". "specific features" or "specific feature selection"?

3. In lines 82-85, "complex interactions may happen within groups but only simple interactions between ... we do not enforce assumptions on the groups such as non-overlapping groups or every feature being...", I do not think it will hold for most cases due to "non-overlapping groups", for example, Group_1=\{x_1,x_2\} and Group_2=\{x_3,x_2\}, if there exist complex interactions between x_1 and x_2, then it will make the interactions between Group_1 and Group_2 become complex. From formula (4), I think it might be more accurate to say that the latent spaces from different groups are simple interactions, right?

4. For the implementation of STG and CAE, the author(s) just simply mentioned "Use the available Implementations". As far as I know, CAE in its original paper involved the use of different structures, such as one with no hidden layer, one with a hidden layer, and so on. So the author(s) should specify which specific structure is used. In addition, for a fair comparison, the compared methods should have similar or same structures.

5. Suggesting that for MNIST visualization examples, please (consider to) give more numerical results, so that the reader can see more clearly which model is really good.

6. At first, I was attracted by the motivation the author(s) demonstrated; that is, the example of genomics at the beginning, but after reading this paper, I was a little disappointed that I did not notice any genome-related data used in the analysis to illustrate the advantages of the proposed method. That's one of the reasons I have to give the current score.

---

In addition, there are some tiny issues/typos:

(1) Journal/conference names are sometimes capitalized and sometimes not.

(2) The author's name in the reference is wrong, such as Ref. [23].

(3) In References, the format of conferences is inconsistent, sometimes abbreviations are given and sometimes not.

(4) In References, the format of authors' names is inconsistent, sometimes abbreviations are used and sometimes not.

I look forward to the response from the author(s), if the response adequately explains and can show the unique superiority of the proposed method in this respect, I will consider raising the score. Thanks.



-----------------------------------------------------------------------

The author(s) clarified most of my concerns. So I would like to raise my score from (4) Borderline reject to (5) Borderline accept. Thanks.

**Limitations:**

Yes.

**Strengths And Weaknesses:**

The topic of this paper looks interesting, I have the following comments:

### Strengths
---
1. The motivation of this paper looks interesting, as the authors illustrated at the beginning that "...in genomics, diseases might not be caused by any single mutation but require the presence of multiple mutations...". This paper attempted to detect groups of predictive features without the need for predefined groups.

2. Experiments on synthetic and semi-synthetic datasets showed some advantages to some extent.

---
---

Although the topic studied by this paper is interesting, I think the paper has the following deficiencies or unclear points (Please see the "Questions" section below for details).

### Weaknesses
---

1. Behind the group feature selection idea proposed in this paper is deep architecture plus ensemble learning, or more simply, ensemble STG in essence. Innovation seems limited.

2. The research review of traditional/standard feature selection methods is not sufficient. In fact, the traditional/standard feature selection methods have been developed even in the past two years, both in terms of method novelty and performance advantages.

3. Maybe the proposed method should be validated on genomics-related data.

---

---

> ### Author Response · Authors · 2022-08-02
> **Response to reviewer LHmD (3/3)**
>
> 5. References: Thank you for bringing this to our attention, we have fixed the references.
>
> __References__
>
> 1. C. Curtis et al. The genomic and transcriptomic architecture of 2,000 breast tumours reveals novel subgroups. _Nature_ 486(7403):346–352, (2012).
> 2. B. Pereira et al. The somatic mutation profiles of 2,433 breast cancers refine their genomic and transcriptomic landscapes. _Nature Communications_ 7(1):11479 (2016).
> 3. M. Bakele et al. An interactive network of elastase, secretases, and PAR-2 protein regulates CXCR1 receptor surface expression on neutrophils. _Journal of Biological Chemistry_, 289(30):20516–20525 (2014).
> 4. B. Chen et al. Differential effects of smad2 and smad3 in regulation of macrophage phenotype and function in the infarcted myocardium. _Journal of Molecular and Cellular Cardiology_, 171:1–15 (2022).
> 5. J. Byerly et al. PRKCQ promotes oncogenic growth and anoikis resistance of a subset of triple-negative breast cancer cells. _Breast Cancer Research_, 18(1):95 (2016).
> 6. C. Leugers et al. Tau in MAPK activation. _Frontiers in Neurology_, 4, (2013).
> 7. W. B. Fang et al. CCL2/CCR2 chemokine signaling coordinates survival and motility of breast cancer cells through smad3 protein- and p42/44 mitogen-activated protein kinase (MAPK)-dependent mechanisms. _Journal of Biological Chemistry_, 287(43):36593–36608 (2012).
> 8. X.-Y. Zhang et al. BMP6 downregulates GDNF expression through SMAD1/5 and ERK1/2 signaling pathways in human granulosa-lutein cells. _Endocrinology_, 159(8):2926–2938, (2018).

---

> ### Author Response · Authors · 2022-08-02
> **Response to reviewer LHmD (2/3)**
>
> __Additional dataset: Real world cancer dataset__
>
> We have tested CompFS on the METABRIC dataset (Curtis et al., 2012; Pereira et al., 2016). METABRIC contains gene expression, mutation, and clinical data for 1,980 primary breast cancer samples. We evaluated the ability to predict the progesterone receptor (PR) status of the tissue based on the gene expression data, which consists of measurements for 489 genes.
>
> Prediction performance:
>
> | Method             | Accuracy | AUC ROC | # Features |
> |--------------------|----------|---------|------------|
> | MLP (All features) | 77.95%   | 0.869   | 489        |
> | CompFS             | 75.07%   | 0.830   | 25         |
> | STG                | 76.12%   | 0.843   | 25         |
> | CAE                | 76.64%   | 0.856   | 25         |
>
> CompFS performs strongly, with limited performance degradation compared to using all features, despite only using 5% of the total features.
> Despite imposing a more rigid structural form on the predictive model and how features can interact, STG and CAE only had marginally greater predictive power than CompFS. However, CompFS provides greater insight into how the features interact (see below).
>
> Qualitative assessment:
>
> We also examined the groups of features selected by CompFS. We found supporting evidence in the scientific literature for all but 1 of the discovered genes (Table 10). In addition, within each group, we found further evidence of the interactions between genes, demonstrating the ability for CompFS to learn informative groups of features.
>
> For example, in Group 1, CXCR1 and PEN-2 (the protein encoded by PSENEN) are known to interact (Bakele et al., 2014).  In Group 2, BMP6 encodes a member of the TGF-$\beta$, and TGF-$\beta$ triggers activation of SMAD3 (Chen et al., 2022). In the same group, MAPK1 activity is dependent on the activity of PRKCQ in breast cancer cells (Byerly et al., 2016), while MAPK1 is also known to interact with MAPT (Leugers et al., 2013), SMAD3 (Fang et al., 2012), and BMP6 (Zhang et al., 2018).
>
> We have included additional supporting evidence in Section 5.3 and Appendix H. Selected features and groups are shown below.
>
> | Group | Features |
> |:----:|:----:|
> | Group 1 | psenen, cxcr1, dlec1, mmp15, srd5a3 |
> | Group 2 | bmp6, mapk1, smad3, mapt, prkcq |
> | Group 3 | bmpr2, mmp12, asxl2, birc6, star |
> | Group 4 | cdkn1a, fgfr1, tgfbr3, npnt, akr1c3 |
> | Group 5 | bmp4, mmp10, tbl1xr1, ush2a, hsd17b1 |
>
>
> __Ranking groups__
>
> Each group selection model is equipped with a group predictor that attempts to solve the supervised learning problem using only the features provided in that group. Using the predictive performance of each group, it is natural to form a ranking of the groups that could be used to prioritize further investigation of features in a real-world use case.
>
> For example, for experiments on MNIST we record these individual group accuracies:
>
> | Group | Accuracy |
> |:----:|:----:|
> | Group 1         | 78.0% |
> | Group 2         | 74.0% |
> | Group 3         | 72.0% |
> | Group 4         | 81.0% |
> | CompFS          | 94.0% |
>
> and for METABRIC we record these individual performances:
>
> | Group | AUC ROC |
> |:----:|:----:|
> | Group 1         | 0.608 |
> | Group 2         | 0.768 |
> | Group 3         | 0.688 |
> | Group 4         | 0.791 |
> | Group 5         | 0.604 |
> | CompFS          | 0.830 |
>
> We see Group 4 is the strongest and Group 3 is the weakest for MNIST, while Group 4 is the strongest and Group 5 is the weakest for METABRIC. We agree that this is not a central point of our paper, and therefore have removed it from the abstract.
>
> __Additional questions__
>
> 1. L46-47: We have revised this sentence as follows:
> >  Discovering _groups_ of features offers deeper insights into _why_ specific features are important than standard feature selection.
> 2. L82-85: We have rewritten this to be more clear. Now in section 3.2 we refer to this as combining the group representations.
> 3. Implementations of STG and CAE: We have provided the exact hyperparameters in Appendix D. STG and the CAE decoder both have two hidden layers with ReLU activations. The hidden widths are 20 and 200. Crucially these methods are not able to find groups, as standard feature selection algorithms.
> 4. MNIST: We have given the results per group and full model in Appendix G, table 9. This is repeated here:
>
> | Group | Accuracy |
> |:----:|:----:|
> | Group 1         | 78.0% |
> | Group 2         | 74.0% |
> | Group 3         | 72.0% |
> | Group 4         | 81.0% |
> | CompFS          | 94.0% |
> | Union           | 95.0% |
> | STG             | 95.0% |
> | CAE             | 95.0% |
>
> Union refers to all features found by CompFS through an MLP, and CompFS refers to the features found by CompFS but using the CompFS architecture. We see that while individual groups perform poorly, when used together they achieve results competitive with features found by STG and CAE, even when used in the highly constrained CompFS architecture.

---

> ### Author Response · Authors · 2022-08-02
> **Response to reviewer LHmD (1/3)**
>
> *Dear reviewer LHmD, thank you for the feedback on our work. We have addressed your comments in the updated version of our manuscript and provide a point-by-point response below.*
>
> __Innovation vs. STG__
>
> Our work has a number of innovations and differences to STG.
>
> First and perhaps the most significant difference, as noted by the reviewer, is that we solve a fundamentally different problem, Composite Feature Selection. We formalize this problem and propose a novel approach to tackle the challenging of _discovering_ groups of predictive features _without_ prior knowledge about feature grouping.
>
> Second, while ensembling is widely used in other areas of machine learning, feature selection methods do not typically use ensembles. We draw parallels between our approach to composite feature selection and ensembling, but note that our approach is not simply a straightforward ensemble of standard feature selection methods.
>
> This leads to another key difference with STG and that is our novel loss function, given in Eqs. (11)-(14). While STG penalizes the sum of the probabilities that gates are active (i.e. an L1 penalization), we use an L2 regularization (Eq. (11)). We explain the rationale for this in L199-206.
> In addition, we introduce a novel regularization term to encourage differentiation between learners (Eq. (13)) to discover distinct groups rather than all learners selecting the same features. Such a term is not used in standard feature selection algorithms, further differentiating our approach.
>
> Finally, as further evidence of the innovation of CompFS compared to STG, we compared CompFS to an ensemble version of STG that concatenates representations of multiple instances of STG (additional model details are given in Appendix G). On most datasets, Ensemble STG performed identically to STG and selected the same features in all members of the ensemble. For Chem3, Ensemble STG found multiple groups of features but did not recover the ground truth group structure, merely two of the learners did not select one of the relevant features. Consequently, for none of the datasets did Ensemble STG discover the ground-truth group structure.
>
> Synthetic datasets:
>
> | Dataset | Method | TPR   | FDR  | G$_{\text{sim}}$ | No. Groups | Accuracy (%) |
> |:-------:|:------:|:-----:|:----:|:-----:|:----------:|:------------:|
> | SYN1    | CompFS       | 100.0 | 0.0  | 0.91  | 2.2        | 98.9         |
> |         | Ensemble STG | 100.0 | 0.0  | 0.50  | 1.0        | 93.5         |
> | SYN2    | CompFS       | 95.0  | 0.0  | 0.90  | 1.8        | 95.5         |
> |         | Ensemble STG | 100.0 | 0.0  | 0.50  | 1.0        | 86.9         |
> | SYN3    | CompFS       | 100.0 | 0.0  | 0.68  | 1.3        | 97.4         |
> |         | Ensemble STG | 100.0 | 0.0  | 0.67  | 1.0        | 84.6         |
> | SYN4    | CompFS       | 90.0  | 51.9 | 0.47  | 2.5        | 95.8         |
> |         | Ensemble STG | 100.0 | 66.7 | 0.17  | 1.0        | 81.2         |
>
> Chemistry datasets:
>
> | Dataset | Method | TPR   | FDR  | G$_{\text{sim}}$ | No. Groups | Accuracy (%) |
> |:-------:|:------:|:-----:|:----:|:-----:|:----------:|:------------:|
> | CHEM1   | CompFS       | 100.0 | 0.0  | 0.82  | 1.9        | 100.0        |
> |         | Ensemble STG | 100.0 | 0.0  | 0.50  | 1.0        | 100.0        |
> | CHEM2   | CompFS       | 100.0 | 0.0  | 0.72  | 2.2        | 100.0        |
> |         | Ensemble STG | 100.0 | 0.0  | 0.50  | 1.0        | 100.0        |
> | CHEM3   | CompFS       | 100.0 | 7.3  | 0.62  | 2.4        | 100.0        |
> |         | Ensemble STG | 100.0 | 0.0  | 0.58  | 2.0        | 99.9         |
>
> __Related work__
>
> We have revised the Related Work section and significantly expanded our discussion of traditional/standard feature selection methods (see Appendix B).
> We agree that there has been significant development in this field recently, and in our experiments compare our approach, CompFS, to recent state-of-the-art methods for standard feature selection STG and CAE, together with an Oracle feature selection method that outperforms _all_ standard feature selection methods.

---

> ### Author Response · Authors · 2022-08-05
> **Dear Reviewer LHmD**
>
> Dear Reviewer LHmD,
>
> Once again, we would like to thank you for your feedback on our work! We hope that our rebuttal has addressed any questions or concerns you may have had about our paper. If you have any other comments or concerns, please let us know - we would be happy to do our utmost to address them during the author-reviewer discussion period (which ends this Tuesday)!
>
> Thank you.

---

> ### Author Response · Authors · 2022-08-07
> **Thank you for your response**
>
> Dear reviewer LHmd,
>
> Thank you for your review and for raising your score. We appreciate the effort in helping improve the paper.
>
> If there are any outstanding points you would like to discuss, we would be more than happy to address them.

---

### Official Review · Reviewer_WTbm · 2022-07-07

**Rating:** 6
**Confidence:** 5
**Soundness:** 2 fair
**Presentation:** 3 good
**Contribution:** 2 fair

**Summary:**

The authors of the paper propose a method for solving the problem of selecting groups of features in a supervised learning setting. The method is based on an ensemble of models where the number of models is the same as the expected number of groups. A new metric is also proposed to evaluate the quality of selected groups of features.

**Questions:**

You claim that there is no source of dataset where the ground truth of groups is provided and that is the reason you use the Oracle method that selects a single group. Is it possible to manually construct such a dataset?

Why do you use a small number of groups for synthetic datasets (up to 3) ? I think it is more convenient to check the method on a larger number of groups and also to evaluate it with varying the number of groups (is its performance degraded)?

How is your method compared to the random initialization of any stochastic gates method? It is possible to train FS net and to get for each initialization different  selected features (assuming some stochasticity in the selection process).

**Limitations:**

The social impact is discussed.
In conclusion the authors describe some common limitations of the feature selection task but it is more reasonable to mention the limitations that are specific to your work: in the definition or in the method.



**Strengths And Weaknesses:**

The paper is written well, it is very clear and is easy to follow. The authors present the challenges of the group selection task, provide formal definition and a simple method to solve the task. The proposed formula for loss seems reasonable.

My main concern is about the problem the authors try to tackle. The incentives which are discussed in the introduction section for solving group feature selection problem are not convincing enough. I think it is better to provide an illustrative example that explains how groups of features could benefit the relevant stakeholders. One of the advantages of feature selection methods is to detect correlated features and exclude them by remaining a single representative for each “correlation group” (e.g. concrete autoencoders). So why should we care about these groups? The correlation could be based on linear and non linear interactions between features. E.g. in yout MNIST experiment, you got groups that look almost the same (1,3,4).
In addition the experimental part where only 3 datasets (Syn1-Syn4 could be labeled as a single dataset)  are used to evaluate the proposed method is not convincing. For datasets selected in the experiments p << number of samples, which is a non challenging setting for supervised feature selection task. I think it is better to justify the significance of your method on the more datasets including the real datasets.
An additional weakness of the proposed method is that it produces probability for each feature to be selected and not a binary vector. It requires an additional hyper parameter to be tuned manually.

Additional technical issues:
Line 67  - w_i is not defined before you mention it
Line 93 -  the space of functions F is not defined before it used here
Line 116 - rho function is not defined / described before it’s used here. If it’s the same as in the paper Deep Sets I think it is better to mention it.

---

> ### Author Response · Authors · 2022-08-02
> **Response to Reviewer WTbm 3/3**
>
> __Limitations__
>
> We have updated the discussion of the limitations of our approach in the conclusion. These are:
> > CompFS is not without its limitations: as with other methods, points of difficulty arise when the input features are highly correlated, or if predictive composites contain overlapping features. This may be overcome in the future by using correlated gates. Further, as with many traditional feature selection methods, there are no guarantees on false discovery rate using CompFS. This could be tackled by using CompFS to propose possible candidate composite features, and then using the Group Knockoff procedure to ensure these guarantees. More broadly, as with standard feature selection, groups found under composite feature selection must be verified by domain experts (both features but additionally interactions).
>
> __Minor comments__
>
> - Line 67 - $w_i$ fixed.
> - Line 93 - the space of functions is not a specific set, for example it can be the space of multilayer-perceptrons of a given width and depth. We have clarified this point.
> - Line 116 - We have clarified this and have now defined $\rho$ and $\phi$. They are the same as those in Deep Sets.
>
> __References__
>
> 1. R. Balendra and A. M. Isaacs. _C9orf72_-mediated ALS and FTD: multiple pathways to disease, _Nature Reviews Neurology_ 14:544–558 (2018).
> 2. D. B. Graham and R. J. Xavier. Pathway paradigms revealed from the genetics of inflammatory bowel disease. _Nature Reviews Neurology_ 14:544–558 (2018).
> 3. M. Kelly and C. Semsarian. Multiple Mutations in Genetic Cardiovascular Disease: A Marker of Disease Severity? _Advances in Genetics, Proteomics, and Metabolomics_ 2:182-190 (2009).
> 4. L. B. Meira et al. Cancer predisposition in mutant mice defective in multiple genetic pathways: uncovering important genetic interactions. _Mutation Research/Fundamental and Molecular Mechanisms of Mutagenesis_ 477(1-2):51-58 (2001).
> 5. J. Merino et al. Polygenic scores, diet quality, and type 2 diabetes risk: An observational study among 35,759 adults from 3 US cohorts. _PLOS Medicine_ 19(4):e1003972 (2022)
> 6. C. Curtis et al. The genomic and transcriptomic architecture of 2,000 breast tumours reveals novel subgroups. _Nature_ 486(7403):346–352, (2012).
> 7. B. Pereira et al. The somatic mutation profiles of 2,433 breast cancers refine their genomic and transcriptomic landscapes. _Nature Communications_ 7(1):11479 (2016).
> 8. M. Bakele et al. An interactive network of elastase, secretases, and PAR-2 protein regulates CXCR1 receptor surface expression on neutrophils. _Journal of Biological Chemistry_, 289(30):20516–20525 (2014).
> 9. B. Chen et al. Differential effects of smad2 and smad3 in regulation of macrophage phenotype and function in the infarcted myocardium. _Journal of Molecular and Cellular Cardiology_, 171:1–15 (2022).
> 10. J. Byerly et al. PRKCQ promotes oncogenic growth and anoikis resistance of a subset of triple-negative breast cancer cells. _Breast Cancer Research_, 18(1):95 (2016).
> 11. C. Leugers et al. Tau in MAPK activation. _Frontiers in Neurology_, 4, (2013).
> 12. W. B. Fang et al. CCL2/CCR2 chemokine signaling coordinates survival and motility of breast cancer cells through smad3 protein- and p42/44 mitogen-activated protein kinase (MAPK)-dependent mechanisms. _Journal of Biological Chemistry_, 287(43):36593–36608 (2012).
> 13. X.-Y. Zhang et al. BMP6 downregulates GDNF expression through SMAD1/5 and ERK1/2 signaling pathways in human granulosa-lutein cells. _Endocrinology_, 159(8):2926–2938, (2018).

---

> > ### Comment · Reviewer_WTbm · 2022-08-07
> > **Thank you for your rebuttal**
> >
> > I have raised my score to 6.

---

> > > ### Author Response · Authors · 2022-08-07
> > > **Thank you for your response**
> > >
> > > Dear reviewer WTbm,
> > >
> > > Thank you for your response and for raising your score. We appreciate the effort in helping improve the paper.
> > >
> > > If there are any outstanding points you would like to discuss, we would be more than happy to address them.

---

> ### Author Response · Authors · 2022-08-02
> **Response to Reviewer WTbm 2/3**
>
> __MNIST__
>
> While perhaps at first glance the Groups appear similar, in actuality they discover quite different components. The best way to see this is in Figure 4 where we show the per-class accuracies of each group.
>
> We see that the performance of each group varies; in particular, Groups 2 and 3 have some digits that are not as accurately classified, while overall CompFS does not. This supports the idea that groups look at different aspects of the problem, and the ensemble combines the information to make an overall prediction.
>
> __Choice of datasets / Use of Oracle__
>
> Both our Synthetic datasets (Syn1-4) and Chemistry datasets (Chem1-3) have known ground-truth group structure. On these datasets, we demonstrate that CompFS is frequently able to recover the group structure and provides additional insights over standard feature selection.
>
> The purpose of the Oracle standard feature selection is as follows: the strongest baseline for the scenarios where the ground truth features are known is Oracle feature selection, which perfectly performs standard feature selection. CompFS outperforms Oracle feature selection at grouping the features. In the revised version of our manuscript, we have included the STG and CAE baselines in the main text. However, by definition, standard feature selection methods cannot outperform Oracle, thus we believe Oracle represents a strong baseline to demonstrate the benefit of group feature selection over standard feature selection.
>
> __Selection probabilities__
>
> The vast majority of feature selection methods do not output a binary vector and instead output a selection probability (e.g. STG, Yamada et al., 2020) or a coefficient that must be thresholded (e.g. LASSO, Tibshirani, 1996).
>
> In addition, we actually believe this is a _strength_ of such methods. Selection probabilities provide a _ranking_ of features, such that one can identify the most important features, rather than all selected features being identified as equally important, which is typically not the case. This allows the user to choose how to include features. In our experiments, we use two approaches, choosing a threshold to select whether a feature is included or not and selecting the top $k$ features.
>
> __Number of groups__
>
> We assessed the impact of the number of groups (among other factors) in an ablation study of our method, shown in Appendix A. In particular, we observe that as the number of possible groups increases, CompFS only uses marginally more of the possible groups (Figure 3). In addition, in all of our experiments, the number of possible groups for CompFS was always set larger than the number of ground-truth groups to more closely simulate real-world use where the true number of groups is unknown.
>
> More generally, as the number of ground truth groups increases, we expect it would be more challenging to recover the entire underlying group structure.
>
>
> __Additional baseline__
>
> As an additional baseline, we compared CompFS to an ensemble version of STG that concatenates representations of multiple instances of STG (additional model details are given in Appendix G). On most datasets, Ensemble STG performed identically to STG and selected the same features in all members of the ensemble. For Chem3, Ensemble STG found multiple groups of features but did not recover the ground truth group structure, merely two of the learners did not select one of the relevant features. Consequently, for none of the datasets did Ensemble STG discover the ground-truth group structure.
>
> Synthetic datasets:
>
> | Dataset | Method | TPR   | FDR  | G$_{\text{sim}}$ | No. Groups | Accuracy (%) |
> |:-------:|:------:|:-----:|:----:|:-----:|:----------:|:------------:|
> | SYN1    | CompFS       | 100.0 | 0.0  | 0.91  | 2.2  | 98.9 |
> |         | Ensemble STG | 100.0 | 0.0  | 0.50  | 1.0 | 93.5  |
> | SYN2    | CompFS       | 95.0  | 0.0  | 0.90  | 1.8  | 95.5  |
> |         | Ensemble STG | 100.0 | 0.0  | 0.50  | 1.0        | 86.9         |
> | SYN3    | CompFS       | 100.0 | 0.0  | 0.68  | 1.3        | 97.4         |
> |         | Ensemble STG | 100.0 | 0.0  | 0.67  | 1.0        | 84.6         |
> | SYN4    | CompFS       | 90.0  | 51.9 | 0.47  | 2.5        | 95.8         |
> |         | Ensemble STG | 100.0 | 66.7 | 0.17  | 1.0        | 81.2         |
>
> Chemistry datasets:
>
> | Dataset | Method | TPR   | FDR  | G$_{\text{sim}}$ | No. Groups | Accuracy (%) |
> |:-------:|:------:|:-----:|:----:|:-----:|:----------:|:------------:|
> | CHEM1   | CompFS       | 100.0 | 0.0  | 0.82  | 1.9        | 100.0        |
> |         | Ensemble STG | 100.0 | 0.0  | 0.50  | 1.0        | 100.0        |
> | CHEM2   | CompFS       | 100.0 | 0.0  | 0.72  | 2.2        | 100.0        |
> |         | Ensemble STG | 100.0 | 0.0  | 0.50  | 1.0        | 100.0        |
> | CHEM3   | CompFS       | 100.0 | 7.3  | 0.62  | 2.4        | 100.0        |
> |         | Ensemble STG | 100.0 | 0.0  | 0.58  | 3.0        | 99.9         |

---

> ### Author Response · Authors · 2022-08-02
> **Response to Reviewer WTbm 1/3**
>
> *Dear reviewer WTbm, thank you for the feedback on our work. We have addressed your comments in the updated version of our manuscript and provide a point-by-point response below.*
>
> __Motivation__
>
> Our central motivation is that in many scenarios (in particular in medicine/biology), individual features do not act alone but together. In addition, often there are multiple groups of variables which act (somewhat) independently of each other. In medicine/biology, these could be different mechanisms or pathways to the outcome. For example, many diseases can manifest from different pathways. Examples include cancer (Meira et al., 2001), amyotrophic lateral sclerosis (ALS) and frontotemporal dementia (FTD) (Balendra and Isaacs, 2018), to inflammatory bowel disease (Graham and Xavier, 2020), cardiovascular disease (Kelly and Semsarian, 2009), and diabetes (Merino et al., 2022).
>
> Standard feature selection can only discover features that are relevant to the target outcome but it __does not__ say _how_ such features are important. In the context of the above examples, standard feature selection will not identify the separate disease pathways.
>
> The purpose of group feature selection is to provide additional information on _how_ features are important to allow researchers to identify and discover such pathways.
>
> To better illustrate how CompFS could be useful, we have added experiments on a real world cancer dataset (see below).
>
> __Additional dataset: Real world cancer dataset__
>
> We have tested CompFS on the METABRIC dataset (Curtis et al., 2012; Pereira et al., 2016). METABRIC contains gene expression, mutation, and clinical data for 1,980 primary breast cancer samples. We evaluated the ability to predict the progesterone receptor (PR) status of the tissue based on the gene expression data, which consists of measurements for 489 genes.
>
> Prediction performance:
>
> | Method             | Accuracy | AUC ROC | # Features |
> |--------------------|----------|---------|------------|
> | MLP (All features) | 77.95%   | 0.869   | 489        |
> | CompFS             | 75.07%   | 0.830   | 25         |
> | STG                | 76.12%   | 0.843   | 25         |
> | CAE                | 76.64%   | 0.856   | 25         |
>
> CompFS performs strongly, with limited performance degradation compared to using all features, despite only using 5% of the total features.
> Despite imposing a more rigid structural form on the predictive model and how features can interact, STG and CAE only had marginally greater predictive power than CompFS. However, CompFS provides greater insight into how the features interact (see below).
>
> Qualitative assessment:
>
> We also examined the groups of features selected by CompFS. We found supporting evidence in the scientific literature for all but 1 of the discovered genes (Table 10). In addition, within each group, we found further evidence of the interactions between genes, demonstrating the ability for CompFS to learn informative groups of features.
>
> For example, in Group 1, CXCR1 and PEN-2 (the protein encoded by PSENEN) are known to interact (Bakele et al., 2014).  In Group 2, BMP6 encodes a member of the TGF-$\beta$, and TGF-$\beta$ triggers activation of SMAD3 (Chen et al., 2022). In the same group, MAPK1 activity is dependent on the activity of PRKCQ in breast cancer cells (Byerly et al., 2016), while MAPK1 is also known to interact with MAPT (Leugers et al., 2013), SMAD3 (Fang et al., 2012), and BMP6 (Zhang et al., 2018).
>
> We have included additional supporting evidence in Section 5.3 and Appendix H. Selected features and groups are shown below.
>
> | Group | Features |
> |:----:|:----:|
> | Group 1 | psenen, cxcr1, dlec1, mmp15, srd5a3 |
> | Group 2 | bmp6, mapk1, smad3, mapt, prkcq |
> | Group 3 | bmpr2, mmp12, asxl2, birc6, star |
> | Group 4 | cdkn1a, fgfr1, tgfbr3, npnt, akr1c3 |
> | Group 5 | bmp4, mmp10, tbl1xr1, ush2a, hsd17b1 |

---

> ### Author Response · Authors · 2022-08-05
> **Dear Reviewer WTbm**
>
> Dear Reviewer WTbm,
>
> Once again, we would like to thank you for your feedback on our work! We hope that our rebuttal has addressed any questions or concerns you may have had about our paper. If you have any other comments or concerns, please let us know - we would be happy to do our utmost to address them during the author-reviewer discussion period (which ends this Tuesday)!
>
> Thank you.

---

### Official Review · Reviewer_jAMg · 2022-07-10

**Rating:** 6
**Confidence:** 5
**Soundness:** 3 good
**Presentation:** 3 good
**Contribution:** 3 good

**Summary:**

The authors address a challenging and important problem of group feature selection. The uniqueness of their work is that they learn the grouping without any predefined knowledge. Feature selection is an important problem in computational biology, and quite often there are groups of features that act together in some biological/physical system. Finding the groups is important for understanding the driving factors behind the system that is being studied. The proposed method for solving this problem relies on an ensemble of feature selection models to find the informative features and groups. The authors demonstrate the effectiveness of their method using synthetic, semi-synthetic, and real data.

**Questions:**

Some suggestions for improving the paper:
-Section 2- Knockoff procedure- > since these works typically have a slightly different goal in mind, it is worth explaining to the reader what are these methods trying to achieve. Just expand with a coupled of lines.
-Section 3- complex vs. simple interactions. These terms have not yet been defined or explained, so this mention is confusing for the reader. Please expand.
-Missing comma after equations 1,2, 3.
-Section 3.2 line 106- when both are known-> do you mean when both are used by the model? The word known may be confusing since the features are always known (they are observed).
-Missing commas after many other equations.
-Why are the results of STG and CAE omitted from the main text? Clearly, these are the strongest competitors, both for the synthetic and for MNIST.
-P9 L327 “c.” not sure what this means.
-Why are you reporting only the average number of selected features? Isn’t the union more informative? Both for interpretability and computational aspects this is more important. Also, how can you compare this to STG and CAE if they select only a union of features? Shouldn’t those be in the same ball park?
I would be glad to consider raising my score if the authors would be able to provide a more challenging example with an evaluation of their method, since I think this is the major weakness of the paper.

**Limitations:**

The authors point out that the limitations appear in the conclusion, but I don’t see a real discussion about the limitations of the method there.

**Strengths And Weaknesses:**

Overall the paper is well written, and I enjoyed reading it. The English level is satisfactory, and the method is motivated well and is scientifically sound.
The proposed method relies on modern tools for feature selection, namely, probabilistic relaxation of Bernoulli variables, which have been found effective for this task in various applications. I would suggest trying a Gaussian distribution instead of the Concrete as shown in STG [40] but this is minor.
One challenge of using the method is tuning the regularization parameters, which might be challenging since there are two in the proposed method.  Another weakness is that the authors don’t use any real-world data (MNIST is not really a dataset for feature selection). The semi-synthetic example is nice, but it seems that multiple methods lead to perfect accuracy, TPR, and FDR.

---

> ### Author Response · Authors · 2022-08-02
> **Response to Reviewer jAMg 2/2**
>
> __MNIST__
>
> _Accuracy_:
> We have rerun our MNIST experiment and rather than thresholding the features, each group selects the top 15 pixels according to the selection probabilities. With two overlapping features this gives 58 features in total, on the same order of magnitude as STG (46) and CAE (45).
>
> The accuracy of the individual groups are given in Table 9 and are as follows:
>
> | Group | Accuracy |
> |:----:|:----:|
> | Group 1      | 78.0% |
> | Group 2   | 74.0% |
> | Group 3     | 72.0% |
> | Group 4      | 81.0% |
> | CompFS      | 94.0% |
> | Union           | 95.0% |
> | STG             | 95.0% |
> | CAE             | 95.0% |
>
> Note that Union refers to an MLP trained on the union of all the features discovered, whereas CompFS refers to a model with the CompFS architecture trained on the discovered groups. We see that while the groups themselves do not perform well, together they are able to achieve the same level of accuracy as state of the art feature selection methods. More importantly, even under the constraints of the CompFS architecture, the level of accuracy remains high, showing the groups found exhibit behaviour consistent with the composite feature definition (Definition 3.1).
>
> In addition, in Figure 4 we show the per-class accuracies of each group. We see that the performance of each group varies and in particular Groups 2 and 3 have some digits that are not as accurately classified, while overall CompFS does not. This further supports the idea that groups look at different aspects of the problem, and the ensemble combines the information to make an overall prediction.
>
> __Union of features__
>
> We have added the total number of features used by CompFS and the other baselines in the MNIST experiment to Table 9.
>
> __Limitations__
>
> We have updated the discussion of the limitations of our approach in the conclusion. These are:
> > CompFS is not without its limitations: as with other methods, points of difficulty arise when the input features are highly correlated, or if predictive composites contain overlapping features. This may be overcome in the future by using correlated gates. Further, as with many traditional feature selection methods, there are no guarantees on false discovery rate using CompFS. This could be tackled by using CompFS to propose possible candidate composite features, and then using the Group Knockoff procedure to ensure these guarantees. More broadly, as with standard feature selection, groups found under composite feature selection must be verified by domain experts (both features but additionally interactions).
>
>
> __Tuning regularization parameters__
> The majority of feature selection methods include a hyperparameter that balances the number of features selected with predictive performance. For example LASSO (Tibshirani, 1996) and STG (Yamada et al., 2018). We use two regularization hyperparameters, one which aims to make groups small, and one which penalizes overlap between groups. This second hyperparameter is required (along with the second term in the loss function) to separate the groups of features such that the groups don't all find the same features. Note that some other feature selection methods also use two or more hyperparameters, for example:
>
> - Elastic Net (Zou and Hastie, 2005) uses two hyperparameters, one for the $L1$ regularization and one for the $L2$ regularization
> - SLOPE (Bogdan et al., 2013) contains as many hyperparameters as features, which can become very large
>
> __Typos__
>
> Thank you for bringing these to our attention. We have fixed these.
>
> __References__
>
> 1. C. Curtis et al. The genomic and transcriptomic architecture of 2,000 breast tumours reveals novel subgroups. _Nature_ 486(7403):346–352, (2012).
> 2. B. Pereira et al. The somatic mutation profiles of 2,433 breast cancers refine their genomic and transcriptomic landscapes. _Nature Communications_ 7(1):11479 (2016).
> 3. M. Bakele et al. An interactive network of elastase, secretases, and PAR-2 protein regulates CXCR1 receptor surface expression on neutrophils. _Journal of Biological Chemistry_, 289(30):20516–20525 (2014).
> 4. B. Chen et al. Differential effects of smad2 and smad3 in regulation of macrophage phenotype and function in the infarcted myocardium. _Journal of Molecular and Cellular Cardiology_, 171:1–15 (2022).
> 5. J. Byerly et al. PRKCQ promotes oncogenic growth and anoikis resistance of a subset of triple-negative breast cancer cells. _Breast Cancer Research_, 18(1):95 (2016).
> 6. C. Leugers et al. Tau in MAPK activation. _Frontiers in Neurology_, 4, (2013).
> 7. W. B. Fang et al. CCL2/CCR2 chemokine signaling coordinates survival and motility of breast cancer cells through smad3 protein- and p42/44 mitogen-activated protein kinase (MAPK)-dependent mechanisms. _Journal of Biological Chemistry_, 287(43):36593–36608 (2012).
> 8. X.-Y. Zhang et al. BMP6 downregulates GDNF expression through SMAD1/5 and ERK1/2 signaling pathways in human granulosa-lutein cells. _Endocrinology_, 159(8):2926–2938, (2018).

---

> ### Author Response · Authors · 2022-08-02
> **Response to Reviewer jAMg 1/2**
>
> *Dear reviewer jAMG, thank you for the positive feedback on our work. We have addressed your comments in the updated version of our manuscript and provide a point-by-point response below.*
>
> __Additional dataset: Real world cancer dataset__
>
> We have tested CompFS on the METABRIC dataset (Curtis et al., 2012; Pereira et al., 2016). METABRIC contains gene expression, mutation, and clinical data for 1,980 primary breast cancer samples. We evaluated the ability to predict the progesterone receptor (PR) status of the tissue based on the gene expression data, which consists of measurements for 489 genes.
>
> Prediction performance:
>
> | Method             | Accuracy | AUC ROC | # Features |
> |--------------------|----------|---------|------------|
> | MLP (All features) | 77.95%   | 0.869   | 489        |
> | CompFS             | 75.07%   | 0.830   | 25         |
> | STG                | 76.12%   | 0.843   | 25         |
> | CAE                | 76.64%   | 0.856   | 25         |
>
> CompFS performs strongly, with limited performance degradation compared to using all features, despite only using 5% of the total features.
> Despite imposing a more rigid structural form on the predictive model and how features can interact, STG and CAE only had marginally greater predictive power than CompFS. However, CompFS provides greater insight into how the features interact (see below).
>
> Qualitative assessment:
>
> We also examined the groups of features selected by CompFS. We found supporting evidence in the scientific literature for all but 1 of the discovered genes (Table 10). In addition, within each group, we found further evidence of the interactions between genes, demonstrating the ability for CompFS to learn informative groups of features.
>
> For example, in Group 1, CXCR1 and PEN-2 (the protein encoded by PSENEN) are known to interact (Bakele et al., 2014).  In Group 2, BMP6 encodes a member of the TGF-$\beta$, and TGF-$\beta$ triggers activation of SMAD3 (Chen et al., 2022). In the same group, MAPK1 activity is dependent on the activity of PRKCQ in breast cancer cells (Byerly et al., 2016), while MAPK1 is also known to interact with MAPT (Leugers et al., 2013), SMAD3 (Fang et al., 2012), and BMP6 (Zhang et al., 2018).
>
> We have included additional supporting evidence in Section 5.3 and Appendix H. Selected features and groups are shown below.
>
> | Group | Features |
> |:----:|:----:|
> | Group 1 | psenen, cxcr1, dlec1, mmp15, srd5a3 |
> | Group 2 | bmp6, mapk1, smad3, mapt, prkcq |
> | Group 3 | bmpr2, mmp12, asxl2, birc6, star |
> | Group 4 | cdkn1a, fgfr1, tgfbr3, npnt, akr1c3 |
> | Group 5 | bmp4, mmp10, tbl1xr1, ush2a, hsd17b1 |
>
> __Knockoff procedure__
>
> We have revised the description of knockoffs to highlight the different aim of such methods as follows:
> >  Finally, the Knockoff procedure is a generative procedure that creates fake covariates (knockoffs), obeying certain symmetries under permutations of real and knockoff features. By subsequently carrying out Feature Selection on the combined real and knockoff data, there are guarantees on the False Discovery Rate. Generalisations of the Knockoff procedure to the group setting also exist, where symmetries under permutations of entire groups must exist.
>
> __Complex vs. simple interactions__
>
> We have updated Section 3 to remove ambiguous terms including "simple" and "complex". We now focus on the predictive model as acting on a set of groups’ latent representations.
>
> __Composite feature selection example__
>
> In this context we do mean known from an observational perspective. The point we wish to make is that if we must observe both $x_1$ and $x_2$ for either to be predictive, then we should group them. This has been changed to clarify it is known by a model.
>
> __Baselines__
>
> The strongest baseline for the scenarios where the ground truth features are known is _Oracle_ feature selection, which perfectly performs standard feature selection. Since CompFS outperforms Oracle feature selection at grouping the features, we omitted standard feature selection methods (which by definition cannot outperform Oracle) from the main text.
>
> In the revised version of our manuscript, we have included the STG and CAE baselines in the main text.

---

> ### Author Response · Authors · 2022-08-05
> **Dear Reviewer jAMg**
>
> Dear Reviewer jAMg,
>
> Once again, we would like to thank you for your feedback on our work! We hope that our rebuttal has addressed any questions or concerns you may have had about our paper. If you have any other comments or concerns, please let us know - we would be happy to do our utmost to address them during the author-reviewer discussion period (which ends this Tuesday)!
>
> Thank you.

---

> ### Author Response · Authors · 2022-08-08
> **Dear Reviewer jAMg**
>
> Dear Reviewer jAMg,
>
> Thank you once again for your positive feedback on our work! We wanted to check in one final time before the end of the author-reviewer discussion period (which ends this Tuesday) to check that our rebuttal has addressed any questions or concerns you may have had about our paper.
>
> In particular, we would like to highlight that we have assessed CompFS on a real-world cancer dataset (METABRIC), both quantitatively and qualitatively validating the discovered features and groups in the scientific literature.
>
> If you have any remaining comments or concerns, please let us know!
>
> Thank you.

---

> > ### Comment · Reviewer_jAMg · 2022-08-08
> > **Response to reviewers**
> >
> > I thank the reviewers for addressing all my concerns. I really like the metabric example, I think this is a valuable addition to the paper. I agree that there are other methods with two regularization parameters, but I think that this is something that needs to be discussed in the text. This could lead to follow-up works that will address this limitation. I still consider the paper a valuable contribution to the community, and I think the proposed method could be useful across many fields.  I am positive about the paper and will keep my score at acceptance, good luck!

---

> > > ### Author Response · Authors · 2022-08-08
> > > **Thank you**
> > >
> > > Thank you for your support! We are very happy that we could address all of your concerns. In particular, we are glad that you liked the METABRIC example!
> > >
> > > Finally, we have added the following sentence to our conclusion to discuss the additional regularization hyperparameter:
> > >
> > > _"Additionally, to discover groups, CompFS requires the introduction of additional hyperparameters which could be challenging to tune in practice."_

---

### Author Response · Authors · 2022-08-02
**Summary of Major Changes**

Dear all reviewers, thank you for your responses. We have incorporated your suggestions and uploaded a revised manuscript, where the changes are made in blue. A pdf of the full paper, including the appendices is included in the supplementary zip file.

More specific details are given below, however here is a brief summary of our changes:

- We ran our method on the METABRIC dataset. METABRIC contains gene expression, mutation, and clinical data for 1,980 primary breast cancer samples. We evaluated the ability to predict the progesterone receptor (PR) status of the tissue based on the gene expression data, which consists of measurements for 489 genes. The results show that our method not just discovers relevant features, but can group them according to their interactions, with the discovered features and interactions supported by the literature.
- We ran four new baselines, which were an Ensemble of STGs, Random Forests, Gradient Boosted Decision Trees and Clustering on Discovered Features. CompFS outperformed all of these approaches at discovering the group structure.
- We have incorporated all other proposed changes to the writing.

---

### Meta-Review · Area_Chair_qK4B · 2022-08-26

**Recommendation:** Accept
**Confidence:** Certain

**Metareview:**

### Summary of paper
This work addresses the problem of grouped feature selection in the supervised learning setting. A new method based on ensemble of features is proposed as well as a new metric to evaluate the results on synthetic, semi-synthetic and real data.

### Rebuttal
Authors were engaged and addressed all the reviewers' concerns and questions.  While reviewers did not engage in the rebuttal discussion nor post-rebuttal discussion, they significantly raised their scores, bringing the average score from 4.75 to 5.75, indicating that their main concerns were addressed.

### Acceptance
There are no major concerns that remain to be addressed and the paper is ready to be published. I hence recommend acceptance of the paper.


**Award:**

No

---

### Decision · Program_Chairs · 2022-09-14

Accept